



# GO_3D_OBS - The Nankai Trough-inspired benchmark geomodel for seismic imaging methods assessment and next generation 3D surveys design (version 1.0)

Andrzej Górszczyk[1,2] and Stéphane Operto[3]

[1]Univ. Grenoble Alpes, ISTerre, 38000 Grenoble, France
[2]Institute of Geophysics, Polish Academy of Sciences, ul. Ks. Janusza 64, 01-452 Warsaw, Poland
[3]Université Cote d'Azur, CNRS, OCA, Géoazur, Valbonne, France, 250 Rue Albert Einstein, 06560 Valbonne

**Correspondence:** Andrzej Górszczyk (agorszczyk@igf.edu.pl)

**Abstract.** Detailed reconstruction of deep crustal targets by seismic methods remains a long-standing challenge. One key to address this challenge is the joint development of new seismic acquisition systems and leading-edge processing techniques. In marine environments, controlled-source seismic surveys at regional scale are typically carried out with sparse arrays of ocean bottom seismometers (OBSs), which provide incomplete and down-sampled subsurface illumination. To assess and minimize

the acquisition footprint in high-resolution imaging process such as full waveform inversion, realistic crustal-scale benchmark models are clearly required.The deficiency of such models prompts us to build one and release it freely to the geophysical community. Here we introduce GO_3D_OBS - a 3D high-resolution geomodel representing a subduction zone, inspired by the geology of the Nankai Trough. The 175 km × 100 km × 30 km model integrates complex geological structures with a visco-elastic isotropic parametrization. It is defined in form of a uniform Cartesian grid containing $33.6e^9$ degrees of freedom

for a grid interval of 25 m. The size of the model raises significant high-performance computing challenges to tackle large-scale forward propagation simulations and related inverse problems. We describe the workflow designed to implement all the model ingredients including 2D structural segments, their projection into the third dimension, stochastic components and physical parametrisation. Various wavefield simulations we present clearly reflect in the seismograms the structural complexity of the model and the footprint of different physical approximations. This benchmark model shall help to optimize the design of next

generation 3D academic surveys - in particular but not only long-offset OBS experiments - to mitigate the acquisition footprint during high-resolution imaging of the deep crust.

## 1   Introduction

To make a step change in our understanding of the geodynamical processes that shape the Earth's crust, we need to improve

our ability to build 3D high-resolution multi-parameter models of deep geological targets at the regional scale. To meet this





challenge, it becomes crucial to design numerical models and related synthetic experiments further promoting a new generation of crustal-scale seismic surveys. Along with the design of new acquisition settings, a quest for high-resolution reconstruction requires us to assess a feasibility of the leading-edge seismic imaging techniques and develop their necessary adaptations to the new acquisition specifications. Among them, tomographic methods including First-Arrival Traveltime Tomography (FAT) (Zelt

and Barton, 1998) and Reflection Tomography (Bishop et al., 1985; Farra and Madariaga, 1988), Slope Tomography (Billette et al., 1998; Lambaré, 2008; Tavakoli F. et al., 2017; Tavakoli F. et al., 2019; Sambolian et al., 2019), Wavefront Tomography (Bauer et al., 2017), Finite-Frequency Traveltime Tomography (Mercerat and Nolet, 2013; Zelt and Chen, 2016), Wave Equation Tomography (Luo and Schuster, 1991; Tong et al., 2014) and Full Waveform Inversion (FWI) (Tarantola, 1984; Mora, 1988; Pratt et al., 1996) shall be examined against large-scale numerical problems and complex synthetic datasets generated in

ultra-long-offset configuration. Combining these methods with the up-to-date seismic acquisition techniques available nowadays should allow for regional-scale seismic imaging of continental margins at sub-wavelength spatial resolution (typically few hundred meters) (Morgan et al., 2013).

During the last decades, a vast majority of deep-crustal seismic experiments carried out by academia were performed in two

dimensions (namely, along profiles), mainly due to the financial cost of such experiments and the lack of suitable equipments. They often combine a short-spread towed-streamer acquisition with a sparse 4C (four-component) OBS (Ocean Bottom Seismometer) deployments. The data recorded by these two types of acquisition conceptually provide complementary information on the subsurface which is processed with different imaging techniques (Górszczyk et al., 2019). Streamer data are used for imaging the reflectivity of the upper-most crust by migration, while the OBS data are routinely utilized to build smooth P-wave

velocity models of the crust by FAT. However, the rapidly increasing inaccuracy of the migration velocities with depth and the lack of aperture coverage provided by streamer acquisition generate poor-quality migrated sections at depths exceeding the streamer length. On the other hand, the low resolution of the velocity models built by FAT prevents an in-depth geological interpretation of complex media. Put together, these two pitfalls currently make the joint interpretation of migrated sections and tomographic velocity models quite illusory. Moreover, off-plane wavefield propagation makes the 2D acquisition geometry

implicitly inaccurate, further increasing the uncertainty of the resulting images. The need for enhancement of this acquisition and processing paradigm is therefore obvious.

Regarding streamer acquisition, up-to-date only a few 3D academic multi-streamer experiments have been conducted (Bangs et al., 2009; Marjanović et al., 2018; Lin et al., 2019). However, these experiments were performed with a small number of

streamers and small spread (typically 6 km). The limited width of the receiver swath leads to narrow azimuth coverage and prevents to survey large area with reasonable acquisition time, while the short length of the streamer limits the maximum depth of investigation as above mentioned. In contrast, the oil industry nowadays carry out 3D wide-azimuth streamer surveys with swath of ∼1 km width and streamers of up to 15 km length arranged in single or multi-vessel configuration. The long streamers and wide receiver swathes combined with shooting the lines in a race track mode would make the acquisition time

of the crustal-scale surveys reasonable (Li et al., 2019) at the same time increasing the penetration depth of the imaging.





For stationary-receiver surveys, coarse 3D passive/active-source OBS experiments begin to emerge in academia (Morgan et al., 2016; Goncharov et al., 2016; Heath et al., 2019; Arai et al., 2019). Sparse areal OBS deployments provide the necessary flexibility to design long-offset acquisition geometries, such that energetic diving waves and post-critical reflections could sample the deepest targeted-structures (lower crust, Moho, upper mantle). While recent trends in exploration industry show intensive
development toward seabed acquisitions carried out with large pools of autonomous nodes (Ni et al., 2019; Blanch et al., 2019), it creates the opportunity to adapt this development to academic regional experiments through a relaxation of the acquisition sampling. Indeed, industry-oriented surveys are often designed with the aim to apply the entire seismic imaging workflow to the recorded dataset (namely velocity analysis and migration). For this purpose, large pools of receivers are required to fulfil sampling criteria for high-frequency migration (Li et al., 2019). This requirement can be relaxed when aiming at the velocity
reconstruction only using frequencies lower than 15 Hz (Mei et al., 2019).

In terms of imaging, processing of streamer data interlaces two tasks under a scale-separation assumption: imaging the reflectivity by depth migration (from ray-based Kirchhoff migration to two-way wave-equation Reverse Time Migration (RTM)) and velocity model building by reflection or slope tomography. Stable velocity model building with those techniques can be
performed down to a maximum depth of the order of the streamer length. In turns, reflectivity imaging by migration may be performed at greater depths if reliable deep velocity information is provided (e.g. from complementary OBS data) and the deep reflection are recorded with an acceptable signal-to-noise ratio. Higher-resolution velocity models can also be built from streamer data by FWI (Shipp and Singh, 2002; Qin and Singh, 2017) if the information carried out by diving waves is made usable after the re-datuming of the data on the seabed (Gras et al., 2019). As an alternative to classical FWI, velocity model
can be built by Reflection Waveform Inversion (RWI), a reformulation of FWI where the reflectivity estimated by least-squares RTM is used as a secondary buried source to update the velocities along the reflection paths connecting the reflectors to the sources and receivers (Xu et al., 2012; Brossier et al., 2015; Zhou et al., 2015; Wu and Alkhalifah, 2015).
All these approaches have been designed to deal with the limited aperture illumination provided by short-spread streamer acquisition, which generates a null space between the low wavenumbers of the velocity macro-model and the high wavenumbers
of the migration image, hence justifying the explicit above-mentioned scale separation during seismic imaging (Claerbout, 1985; Jannane et al., 1989; Neves and Singh, 1996). Compared to streamer data, long-offset OBS data recorded by stationary-receiver surveys contain a wider aperture content and a richer information about the deep crustal structures. This information can be processed by traveltime tomography and waveform inversion techniques - the former providing a kinematically-accurate starting model for the latter (Kamei et al., 2012; Górszczyk et al., 2017). By long-offset data it is meant a seismic data which
record diving and refracted waves that undershoot the deepest targeted structures. Under this condition, the angular illumination of the structure is sufficiently wide to build high-resolution crustal models by FWI. Therefore, FWI should be considered as the method of choice for long-offset stationary-receiver data.

FWI is a brute-force nonlinear waveform matching procedure which allows for the building a broadband subsurface models
- provided that the structure is illuminated by a variety of waves propagating from the transmission to the reflection regime





(Sirgue and Pratt, 2004). Diving waves, pre- and post-critical reflections, diffractions etc. have potential to generate a wide range of scattering from the targeted structure and therefore can probe this structure with various wavenumber vectors (Figure 1(a)). By broadband model we understand a low-pass filtered version of the true earth, where the local cut-off wavenumbers in each $x,y,z$ spatial direction are controlled by the local wavelength and the scattering, dip and azimuth angles sampled by the

acquisition (Figure 1(b)). This means that the resolution obtained with FWI depends not only on the frequency bandwidth of the data, but also on the aperture with which the wave interacts with the heterogeneities to be reconstructed (Operto et al., 2015). Fulfilling this wide-angular illumination specification is the first fundamental methodological issue for successful application of FWI to OBS data. Indeed, this wide aperture illumination provided by long-offset acquisition is achieved at the expense of the receiver sampling, whose imprint in the imaging should be assessed. When a limited pool of instruments is available,

the down-sampling of the acquisition is directly mapped into the down-sampling of the subsurface wavenumbers, leading to spatial aliasing or wraparound in the spatial domain. A second fundamental methodological issue is therefore to mitigate the aliasing artefacts by reliable compressive sensing techniques and sparsity-promoting regularization during FWI (Herrmann, 2010; Aghamiry et al., 2019b). The third main issue is related to the well-known nonlinearity of the FWI associated with cycle skipping (e.g. Virieux and Operto, 2009). When a classical form of the least-squares misfit function is used (namely, the least-

squares norm of the difference between the simulated and the recorded data), FWI can remain stuck in a local minimum when the initial model does not allow one to predict traveltimes with errors smaller than half a period. The cycle-skipping condition is indeed increasingly difficult to satisfy when the number of propagated wavelengths increases as in case of ultra-long-offset regional surveys (Pratt, 2008). This issue can be addressed by developing kinematically-accurate starting velocity models by traveltime tomography (Górszczyk et al., 2017), by breaking down FWI into several data-driven multiscale steps according to

frequency, traveltime and offset continuation strategies (Kamei et al., 2013; Górszczyk et al., 2017), by designing more robust distances in FWI (Warner and Guasch, 2016; Métivier et al., 2018) or by extending the linear regime of FWI by relaxation of the physical constraints (van Leeuwen and Herrmann, 2013; Aghamiry et al., 2019a). The final key challenge is related to the computational burden resulting from wave simulation in huge computational domains. The only panacea here can be given through a development of efficient, massively-parallel modelling, inversion and optimization schemes tailored for the

high-performance-computing architecture available nowadays.

As reviewed above, advanced deep-crustal seismic imaging raises different frontier methodological issues in terms of acquisition design, inverse problem theory and high-performance computing. Addressing these challenges, can be fostered by establishing geologically meaningful 3D marine regional model amenable to explore pros and cons of different acquisition ge-

ometries, wave propagation requirements and imaging techniques. Indeed, most of the geomodels in the imaging community have been designed to fit the specifications of the exploration scale (Marmousi (Versteeg, 1994), SEG/EAGE salt and overthrust (Aminzadeh et al., 1997), 2004 BP salt model (Billette and Brandsberg-Dahl, 2004), SEAM models (Pangman, 2007)), while available crustal-scale geomodels are rather smooth or lack well defined structures and are 2D (e.g. 2D CCSS blind-test model (Hole et al., 2005; Brenders and Pratt, 2007b, a)). In contrast our proposed geomodel shall incorporate a representative

sample of three-dimensional geological structures detectable at the seismic scale (faults, sediments cover, main structural units





and discontinuities) and a multi-parameter physical definition of those structures (Wellmann and Caumon, 2018). A suitable natural environment to fulfil such specifications is provided by subduction zones. Indeed, the complex geological architecture of these margins warrants the variety of structures characterized by distinct physical parameters. Moreover, convergent margins still crystallise many studies on the role of structural factors (sea mounts, subduction channel etc.) and fluids on the rupture
process of megathrust earthquakes (Kodaira et al., 2002). Therefore, capitalizing our experiences from the previous imaging studies in the region of the eastern Nankai Trough (Dessa et al., 2004; Operto et al., 2006; Górszczyk et al., 2017, 2019) and combining them with the diverse geological features - typically interpreted in the convergent margins around the world - we aim at building a regional synthetic model of subduction zone.

The proposed geomodel is intended to serve as a experimental setting for various imaging approaches with a special emphasise on multi-parameter waveform inversion techniques. For this purpose, the size of geological features we introduce shall be detectable by seismic waves and span from tens of kilometres (major structural units building mantle, crust, volcanic ridges etc.) to tens of meters - namely to the order of smallest seismic wavelengths (sedimentary cover, subducting channel, thrusts and faults etc.). The structural complexity of the model is complemented by a broad range of physical parameters.
Our approach incorporates deterministic, stochastic and empirical components at various stages of the model-building. The deterministic components cover the shape of the main geological units, as well as their projection into the third dimension. The stochastic components include small scale perturbations and random structure warping - introducing further spatial variation (Holliger et al., 1993; Goff and Holliger, 2003; Hale, 2013). The empirical components impose the physical parametrisation - $V_p$, $V_s$, $\rho$ (density), $Q_p$, $Q_s$ - in terms of the magnitude of subsequent parameters and relations between them (Brocher, 2005;
Wiggens et al., 1978; Zhang and Stewart, 2008). Such a combination of structural and parametric variability is reflected by the anatomy of the seismic wavefield, making it suitable to benchmark different imaging approaches. We obtain the 3D cube through the projection of the initial 2D inline profile towards the strike direction of subduction, leading to realistic structure variations along the crossline extension of the model. The dimension of the final model equals to 175 km $\times$ 100 km $\times$ 30 km leading to 33.6e$^9$ degrees of freedom in a Cartesian grid with a 25 m grid step. Therefore, apart from the challenges related
to the seismic imaging of the geologically complex setting, the model imposes significant computational burden in terms of seismic wavefield modelling. The large size of the computing domain can further contribute to the development of efficient forward/inverse problem solvers (time- frequency-domain wavefield modelling, eikonal and ray-tracing solvers etc.) dedicated for crustal-scale imaging.

This article is organised as follow. We start with the description of the geological units which build the 2D curvilinear structural
skeleton of the model. Then, we review how we project the 2D initial structure into the third dimension and how we introduce the visco-elastic properties in each structural units. Third, we describe the implementation of the stochastic components designed to introduce more structural heterogeneity. Finally, we present some simulations of OBS data in the 3D model with the finite difference and spectral element methods and we summarize the article with discussion and conclusion.





## 2 Model building

In the following sections we present step by step the workflow that was designed to create the seismologically representative model. The whole procedure was implemented in MATLAB environment.

### 2.1 Geological features

The overall geological setup of our model is mainly (but not only) inspired by the features which were interpreted in the Nankai Trough area. However, these structures can be also found in different margins around the world combined in various

configurations. Therefore, our model is not intended to replicate a particular subduction zone and its related geology for geodynamic studies of the targeted region. On the contrary, it was design to have broad features one may encounter in these tectonic environments. Such quantitative subsurface description should challenge performances of high-resolution crustal-scale imaging from the complex waveforms generated in this model.

The skeleton of the initial 2D model is presented in Figure 2(a). It is designed as a cross-section aligned with the direction

of subduction and consists of 46 geological units. These units are defined by means of the interfaces creating independent polygons and obtained in the following three steps. First, we pick the spatial position of the nodes in the cross-section (see dots in Figure 2(a)). Second, we interpolate piecewise cubic Hermite polynomials through a subset of nodes to build a boundary (or interface) between the geological units. Such a boundaries represent either lithostratigraphic discontinuities or tectonic features like faults. They can connect and/or intersect each other to create closed regions in the (x,z) plane representing separate

geological units. We refer these closed regions to as polygons. With such an approach we can modify the shape of existing or add new features to the initial structure.

To create an overall skeleton of the model, we start from the interfaces that shape the mantle and the crust. In Figure 2(a), the top of the oceanic/continental crust and mantle are marked by solid red and green lines, respectively. The geometry of the subducting slab is inferred from previous seismic imaging results in the Tokai area. The oceanic crust exhibits variability of

thickness and local bending consistently with what is proposed by geological studies in the same region (Kodaira et al., 2003; Le Pichon et al., 1996; Lallemand et al., 1992; Górszczyk et al., 2019) and addressed as an effect of subducting volcanic ridges and sea mounts. On the right flank of the model (starting at 140 km of model distance) we introduce a large volcanic ridge indicated by the uplift of the bathymetry and the significant thickening of the oceanic crust. Similar ridge (Zenisu Ridge) really exists in the eastern part of the Nankai Trough. It currently approaches the subduction zone and simultaneously undergoes the

thrusting process (Lallemant et al., 1989).

In the next step we cut the subducting oceanic crust and mantle by series of interfaces (blue dashed lines in Figure 2(a)) to subdivide them into several polygons. These interfaces allow us to project each of the defined block independently in the third dimension (see next section). They are also designed to implement tectonic discontinuities in the oceanic crust and upper mantle. Such discontinuities acting as faults, thrusts or creating horst-graben structures are well documented in subduction

zones (Tsuji et al., 2013; Ranero et al., 2005; Azuma et al., 2017; Boston et al., 2014; Vannucchi et al., 2012; von Huene et al., 2004). At the edge of the continental part, two blue dashed lines mark major boundaries creating large flower-like structures,





which can be typically found around strike slip faults zones (Huang and Liu, 2017; Ben-Zion and Sammis, 2003; Tsuji et al., 2014). Each of those boundaries can be also further used to implement fluid paths or damage zones.

On top of the oceanic crust we place two layers of subducting sediments (grey dashed and solid lines in Figure 2(a)). They
extend starting from the subducting channel up to the right edge of the model. In general the lower layer is thicker and overlay directly the top of the oceanic crust covering its thrusts and faults. The upper layer is thinner, whereas its upper limit delineates the decollement in our model. To provide more complexity, we introduce a small duplexing structure within those two layers (between 65 km and 85 km in Figure 2(a)) similarly to the decollement model presented by Kameda et al. (2017), Hashimoto and Kimura (1999) and Collot et al. (2011).

Just between the edge of the continental crust and the subducting slab, we add a stack of relatively thin sheets corresponding to progressively underplated material (dashed black lines in Figure 2(a)). Typically these kind of structures occur when sediments (which move down on top of the subducting oceanic crust) create a duplex system and are further added to the accretionary wedge (Angiboust et al., 2014, 2016; Menant et al., 2018; Agard et al., 2018; Ducea and Chapman, 2018). Including these relatively fine scale structures in the deep part of the model shall significantly affect wavefield propagation.

The accretionary wedge consists of two parts. The land-ward part is formed by large deformed stacked thrusts sheets (solid blue lines in Figure 2(a)) corresponding to the old accretionary prism which extends sea-ward into large out-of-sequence thrusts acting as a backstop now. Similar inner-wedge structures can be found in various subduction zones (Dessa et al., 2004; Raimbourg et al., 2014; Shiraishi et al., 2019; Collot et al., 2008; Cawood et al., 2009; Contreras-Reyes et al., 2010), as well as modelled with analogue sand-box experiments (Gutscher et al., 1998, 1996). The outer-wedge (red shaded polygon in Figure
2(a)) contains the sedimentary prism adapted from Kington and Tobin (2011) and Kington (2012) (Figure 2(b)). It is built out of different types of sediments which underwent deformation and created a sequence of thrusts. The complex-shaped, small-scale, steeply-dipping structures shall impose a challenge for seismic imaging. In our implementation the prism from Figure 2(b) is managed as a single block associated with the respective red shaded polygon in Figure 2(a). Depending on the changes of the shape of this polygon during projection step, we re-sample the prism from Figure 2(b) such that it conforms to the shape
of the polygon.

Finally, the frontal prism is fed by the layers of incoming sediments (black lines in Figure 2(a)) - relatively thick in the area of trench and thin over the volcanic ridge. These sedimentary layers are associated to those interpreted in Figure 2(b). Similarly, on the land-ward part of the model we implement a sedimentary cover on the backstop including the fore-arc basin over the old part of the prism (between 40 km an 70 km). The whole model is covered with a water layer with strongly variable bathymetry.

## 2.2   Projection

The next step in our scheme is the geologically guided projection of the initial skeleton (Figure 2(a)) into the third dimension. We design the set of projection functions which translate the initial inline skeleton in the crossline direction, such that the resulting structure follows concepts related to geology variations along strike direction. For each new inline section, the $(x(y_0), z(y_0))$ coordinates of the initial node (dot in Figure 2(a)) are firstly projected in the crossline direction with an arbitrarily-chosen de-
terministic functions of the crossline distance $y$. In this way for a given inline $i$ we obtain a new node with the coordinates



$(x(y_i), z(y_i))$. The general formula of the projection functions can be expressed as a parametric equation of the following form:

$$
\begin{cases}
x(y_i) = a y_0^2 + b y_0 + c \\
z(y_i) = d y_0 + e.
\end{cases}
\tag{1}
$$

It is immediate to see that $x(y)$ and $z(y)$ are quadric and linear functions respectively, which control curvature and dipping of

a projected structure with the crossline distance $y$.

Once the nodes defining a given interface are projected in the crossline direction, we interpolate the piecewise cubic polynomials between the new nodes in the inline direction. In this way we create a new interface which is slightly shifted with respect to that of the previous inline section. The functions used to project each of the node from the sub-set of nodes defining a given interface are designed such that they are similar but not exactly the same. This means that the interface is not only shifted (as if

exactly the same projection function was used for all the nodes defining the interface) but its geometry can be also modified. In Figure 3(a) we show the main fault planes of the model obtained during projection of the nodes (dots extracted at each 5 km of crossline distance) defining the corresponding interfaces. The figure highlights how the dipping and bending of the presented surfaces, as well as their spatial extension in the $(x, z)$ planes, vary in the crossline direction. From this, it is clear that based on geometrical constraints (which may describe the complex geological evolution) we are able to modify the shape of the

polygons as we project the interfaces from one inline section to the next. The position of the nodes of a given interface can also be defined according to the position of the nodes of an another interface. For example, the nodes discretizing the interface marking the top of the subducting sediments (Figure 2(a), grey dashed and solid lines) can be easily defined (shifted up) once we know the coordinates of the nodes defining the top of the oceanic crust (Figure 2(a), red line).

It is worth mentioning that some nodes can belong to several interfaces - for example, when they are at the junction of two

interfaces. In such a case, these nodes are projected only once with the function assigned to the first interface in order to guarantee a conform space warp of the polygons. Therefore, the projection function applied to the second interface must be partially determined by the fact that one of the nodes (which belongs also to the first interface) was already projected. In such a way, the structural skeleton we create can be viewed as a system of interfaces which are implicitly coupled via the dependencies between the position of the nodes which define them. Therefore, adding new features to the model or modifying existing

ones requires careful design of the corresponding projection functions which will honour the previous dependencies between interfaces.

We assumed that the model will be 100-km width, therefore we generate 4001 2D inline sections spaced 25 m apart. To highlight the lateral variability of the structure generated during projection, we show in a perspective view the skeleton of the inline sections extracted every 20 km (Figure 3(b)). First, we design the curved shape of the subduction front. With increasing

crossline distance, the oceanic crust and mantle shrinks-back. As a result, the continental part of the mantle and crust elongates seaward. Simultaneously, we compress and extend the respective polygons making their relative size variable. One can track this variability by following the geometry of the interfaces cutting through the oceanic mantle and crust along successive inline sections (blue dashed lines in Figure 3(b)). The absolute depth of the oceanic crust is not only changing along the subduction





trend but also gently increasing with crossline distance. Simultaneously, there is an up-growing of the underplating sediments
below the edge of the continental crust (black dashed lines in Figure 3(b)) - consistent with the thickening of the subducting
sediments as the crossline distance increases. The position of the backstop is also changing from one inline to another. This
partially results from the curved shape of the subduction front and was inspired by the 3D interpretation of Tsuji et al. (2017).
At the same time, the length of the inner wedge (solid blue lines) increases. This is coupled with the differences in scale and
deformation of the thrusts sheets which are building this unit, as well as with the shape of the forearc basin covering it. Further
seaward, the outer wedge (red shaded polygon) becomes shorter and thicker as the crossline distance is increasing. The right
flank of the model is marked by the uplift of the bathymetry related to the presence of the incoming volcanic ridge which
becomes smaller as the crossline distance increases.

In Figure 3(c-f), the crossline sections extracted at 40 km, 70 km, 100 km, 130 km (vertical red dashed lines in Figure 2(a))
highlight the structural heterogeneity in the crossline direction. This heterogeneity is defined at different scales. Firstly, due
to the overall curved shape of the subducting front. Secondly, due to the different projection functions causing warping of the
polygons. The structural complexity will be further increased through parametrization and application of stochastic compo-
nents.

## 2.3 Parametrization

### 2.3.1 Index and gradient matrix

Once the inline section has been projected in the crossline direction, the newly generated structural skeletons are further
mapped into uniform 2D Cartesian grids of dimension $1201 \times 7001$ (grid size 25 m $\times$ 25 m). During this step, we assign an
arbitrary index to the grid points representing different geological units. Namely, the grid points inside each of the polygons
are now described by a unique integer value. By re-sampling in the horizontal and vertical directions the matrix discretizing
the accretionary block shown in Figure 2(b), we reshape the geometry of the outer wedge such that it matches that of the
corresponding polygon (red shaded area in Figure 2(a)) in each projected inline section. The deformed sediment layers inside
the outer wedge have indexes consistent with those of the sediments layers incoming into the trench since they represent the
same geological formations. Filling all the polygons with the corresponding indexes leads to the matrix plotted in Figure 4(a).
We pick random colours to distinguish between the neighbouring units. The matrix contains 46 geological blocks defined by
the unique integer index. Such indexation allows us to easily refer to a particular structural unit of the model during the further
steps - for example to modify the parametrization.

On top of the index matrix, we implement another matrix, referred to as gradient matrix, which allows to introduce spatial
variations of the physical parameters in each large-scale unit and around main fault planes (Figure 4(b)). Without this second
matrix, the physical parameters in each geological unit would have constant value (related to the integer from matrix of in-
dexes). The spatial variation of the parameters within the same unit can be related to increasing depth in the mantle, layering
of the crust, low-velocity zones in the subducting sediments, compaction in the prism or damage zones around the faults etc.
We implement those variations using horizontal and/or vertical gradient functions (i.e., linear interpolation functions). These





functions can be easily modified and hence provide the necessary flexibility to update the variations of the physical parameters in the structural units. The gradients in Figure 4(b) contain float values which are normalized between 0 and 1 for each unit. The exceptions are the three units in the oceanic mantle (which do not reach bottom of the model), as well as shallow sedimentary

layers and water column (no gradient implementation).

In practice, the index matrix and the gradient matrix are combined to implement any physical parameter in each geological formation. To do so, we extract a given geological unit (based on the index matrix) and assign to it a constant parameter value $p_\alpha$. We also extract the corresponding part of the gradient matrix and denote it by $\mathbf{G}_n$ (normalized gradient matrix). The gradient function associated with $\mathbf{G}_n$ can be manipulated (scaled, clipped, biased etc.) leading to the matrix $\mathbf{G}$ which provides a

desired spatial variation of a parameter in a geological unit. The overall formula to compute the physical parameters inside a given model unit is as follow:

$$p(x,z) = p_\alpha + p_\beta g(x,z), \tag{2}$$

where $p(x,z)$ is the parameter entry at the position $(x,z)$, while $p_\beta$ scales the entry $g(x,z)$ of $\mathbf{G}$ with appropriate physical units.

To better illustrate how the index and gradient matrix can be used to produce a physical model, we consider the following example. Let us imagine that we want to assign $V_p$ values inside the part of the oceanic crust delineated by the thick red line in Figure 4(a-b). The integer index assigned to this unit is 11 (Figure 4(a)). We set $p_\alpha$ to 4800 m/s and $p_\beta$ to 2900 m/s. Taking into account that the values of $\mathbf{G}_n$ for this unit span from 0.0 to 1.0 (from the top to the bottom of the crust) the final $V_p$ values will be increasing linearly from 4800 m/s to 7700 m/s. One may also desire to implement a gradient discontinuity in the oceanic

crust corresponding to the oceanic layer 2-layer 3 boundary. Setting the thickness ratio to 0.3 and 0.7 in the layer 2 and layer 3, respectively, would split $\mathbf{G}_n$ in unit 11 into two sub-matrices given by:

$$\begin{cases} \mathbf{G}_1 = \mathbf{G}_n * 3.(3); & 0.0 \le \mathbf{G}_n < 0.3 \\ \mathbf{G}_2 = (\mathbf{G}_n - 0.3)/0.7; & 0.3 \le \mathbf{G}_n < 1.0 \end{cases} \tag{3}$$

As a result we obtain two gradient matrices $\mathbf{G}_1$ and $\mathbf{G}_2$ with normalised values between 0.0 and 1.0. We can now set the $p_\alpha$ to 4800 m/s and $p_\beta$ to 1800 m/s for $\mathbf{G}_1$. As a result we obtain rapid increase of $V_p$ between 4800 m/s to 6600 m/s inside layer 2.

For $\mathbf{G}_2$ we use $p_\alpha$ equal to 6800 m/s and $p_\beta$ equal to 1000 m/s leading to $V_p$ values inside layer 3 ranging between 6800 m/s and 7800 m/s. The resulting boundary between layer 2 and layer 3 is marked with dashed line in Figure 4(c).

We also use the gradient matrix to implement variations of the parameters within the subducting sediments, damage zones and subducting channel which result from a small-scale geological processes (fluid migrations, lithological variations, tectonic compaction, mass wasting etc., (Agudelo et al., 2009; Park et al., 2010). Moreover, we implement lateral parameter disconti-

nuities at the interfaces acting as faults. For example, the stairs in the Moho resulting from the interfaces cutting through the oceanic mantle and crust imply that the gradients on both sides of those interfaces are slightly different. In consequence we create a lateral jump in the parameter values on both side of the fault. This jump is in practice more pronounced near to the top of the mantle ($\sim$5 km to $\sim$7 km below Moho) and is vanishing with increasing depth to completely disappear at the bottom of





the model.

Finally, the gradient matrix describes also two additional types of perturbations. The first type is implemented around the regions of thickened oceanic crust mimicking subducting volcanic rides (Park et al., 2004; Kodaira, 2000). They are marked by the gradient and velocity variations in Figure 4(b-c) (95 km - 105 km and 145 km - 155 km). The second type of perturbations is intended to introduce more heterogeneity around the fault planes such as damage zones or fluid paths. Parameter variations within such zones affect the kinematic and dynamic characteristic of the propagating wavefield. They also have potential to
generate distinct arrivals - so-called trapped waves (Li and Malin, 2008; Ben-Zion and Sammis, 2003). We implement such anomalies around the faults in the oceanic crust and upper mantle, the major faults at the edge of continental crust, as well as within the large thrusts sheets between inner and outer accretionary wedge. Few small perturbations are also locally added between layers of inner wedge to increase the complexity of this unit.

**2.3.2 Physical parameters**

From the index and gradient matrices described in the previous section, we can now implement the seismic properties in the structural units. In practice, this requires some constraints on the parameters, which might come from field experiments and laboratory measurements. Regional seismic studies often lack resolution to directly map the reconstructed properties into the fine-scale structures of our model. Moreover, they often ignore second-order parameters such as s-wave velocity ($V_s$), density
($\rho$), attenuation ($Q_p$ and $Q_p$) and anisotropy to focus on the p-wave velocity ($V_p$) reconstruction. Therefore, we first develop a $V_p$ model from recent FWI case studies in the Nankai trough. Then, we use this $V_p$ model as a reference to build the other parameters - namely $V_s$, $\rho$, $Q_p$, $Q_s$ - from empirical relations.

Reconstruction of $V_p$ from wide-angle seismic data has long historical records (Christensen and Mooney, 1995; Mooney et al., 1998). Traveltime tomography has produced a vast catalogue of smooth $V_p$ models from different regions around the world.
This information (even though values can significantly vary depending on the studied area) is useful to constrain the velocity trend in the main large-scale geological units (such as mantle, continental and oceanic crust). Moreover, recent OBS FWI case studies partially fill this resolution gap (Kamei et al., 2012; Górszczyk et al., 2017) and allow us to refine $V_p$ in the short-scale units of our model (Figure 4(c)). We perform this refining of the $V_p$ velocities by trial-and-error, until the OBS gathers simulated under acoustic approximation in the $V_p$ model exhibit in overall a similar anatomy to the OBS gathers collected in the
eastern-Nankai trough (Górszczyk et al., 2017).

From the $V_p$ model, we build $V_s$ and $\rho$ using the empirical polynomial relations of Brocher (2005). These relations were inferred from compiled data on both laboratory measurements, logs, vertical seismic profiling and tomographic studies. They are relevant for $V_p$ ranging from 1500 m/s to 8500 m/s, and hence represent the average behaviour of crustal rocks over the depth range covered by our model. The resulting $V_s$ and $\rho$ models are presented in Figure 5(a-b). Shear-wave velocities in the
shallow sediments are as small as $\sim 530$ m/s. Although even lower velocities can be found in the first few meters below seabed, these values already impose significant challenges for wavefield modelling. In the oceanic crust, $V_s$ increases from 3200 m/s to 4300 m/s, while it starts at 4600 m/s on top of the upper mantle and gently increases with depth. Density of the oceanic and





continental crust varies from 2600 kg/m$^3$ to 3200 kg/m$^3$ and from 2500 kg/m$^3$ to 2900 kg/m$^3$ respectively, while in the upper mantle $\rho$ starts around 3200 kg/m$^3$. Figure 5c shows $V_p/V_s$ ratio, which varies from 1.6 to 3.0. Initially, the $V_s$ model obtained

from empirical relations of Brocher (2005) was not producing heterogeneous $V_p/V_s$ ratio in the subduction channel - as it can occur when fluid overpressure, fluid diffusion or hydro-geologically isolated zones generate variation of the seismic properties in the subduction channel (Kodaira et al., 2002; Collot et al., 2008; Ribodetti et al., 2011). Therefore, we additionally rescale the velocities in this unit using the index and gradient matrices.

We also implement attenuation effects in our model through the $Q_p$ and $Q_s$ parameters. The choice of consistent approach to

constrain plausible $Q_p$ and $Q_s$ models is quite challenging. This is firstly because the various types of attenuation (intrinsic and scattering-related) can be controlled by countless factors - rock type and mineralogy, porosity, fluid content, saturation etc. Secondly, values of $Q$ factor can differ significantly between studies depending on the used methodology, scale of the attenuation or frequency content of seismic data. Thirdly, accurate reconstruction of $Q$ models from field experiments is not so well documented because the footprint of attenuation in the wavefield remains small in particular in the deep crust. Therefore,

to build the $Q_p$ and $Q_s$ models, we combine few empirical relations between velocity ($V_p$, $V_s$, $V_p/V_s$) and attenuation ($Q_p$, $Q_s$) which we find consistent for our synthetic model. To build the $Q_s$ model we use the following power law $Q_s=0.0053V_s^{1.25}$ (Wiggens et al., 1978). This empirical relation is in good agreement with the $Q_s$ estimation of (Olsen et al., 2003) performed in the Los Angeles Basin. The estimated $Q_s$ values range from ∼25 in the shallowest sediments to ∼220 in the deep mantle (see Figure 5(e)). To avoid constant ratio between $Q_p$ and $Q_s$ ($Q_p=1.5Q_s$ in Olsen et al. (2003)), we generated $Q_p$ model

(Figure 5(d)) via another empirical relation $Q_p=36.8(V_p)$-49.6 - derived from well-log data by Zhang and Stewart (2008). This formula, even though derived from velocities up to ∼3000 m/s using sonic-log frequency band, produces reasonable $Q_p$ variations between ∼35 and ∼260 in our model, and leads to $Q_p/Q_s$ ratio ranging from ∼1.1 to ∼1.5 (Figure 5(f)), which is in good agreement with the values estimated in the Mariana subduction zone (Pozgay et al., 2009). The small $Q$ values in the shallow sediment layers and/or subducting channel shall significantly affect the short to intermediate offset wavefield. On the other

hand, the higher $Q$ values in the crust and mantle can still affect the ultra-long offset waves penetrating for a long time through these parts of the model. The $Q$ values we define here are related to the intrinsic attenuation during the wavefield propagation. Additionally, the attenuation of high-frequency wavefield component through the scattering effects is introduced in our model by the means of small scale stochastic perturbations as described in the following section.

## 2.4 Stochastic components

In the final step of our model-building procedure, we consider various types of stochastic components, which are intended to introduce perturbations of random character into the model. We design two types of such a random components: (i) small scale parameter perturbations, and (ii) spatial warping of the final 3D cube.





### 2.4.1  Small scale perturbations

In Figure 4(c) and Figure 5, the parameters are varying smoothly within the units according to the applied gradient functions. Indeed, such smooth variations are rather idealised vision of the true earth. Therefore, to make it more realistic, we create an another matrix describing small scale perturbations (Figure 6(a)) which shall have a second order impact on the wavefield propagation. Using this matrix, we can scale the models after parametrization and obtain the updated model presented in Figure 6(b) (compare with the smooth velocity variations of the $V_p$ model shown in Figure 4(c)). The matrix in Figure 6(a) is obtained by stacking 3D disk-shaped structural elements (SE) (Figure 6(c-f)). The values in the SE (Figure 6(g)) vary from 0 at the edge to 1 at the centre and are further translated into parameter perturbations. We can control the position and size of these SE, as well as their correlation lengths ($z$,$x$,$y$,) - depending on a desired characteristic of the final perturbations.

In Figure 6(c-f), we show four different 3D stacks (10 km × 10 km × 10 km sub-volumes are displayed), corresponding to four different scales of the SE with correlation lengths ratio equal to 0.25 × 1 × 1. The red/blue colours correspond to the positive/negative amplitudes, which are randomly assigned to each of the SE during stacking. These random variations of positive/negative amplitudes guarantee that the final distribution of the stochastic perturbations for each geological unit in the matrix shown in Figure 6(a) has a zero mean. The position of the SE within the stack is controlled by the Cartesian coordinates. The distances between the centres of neighbouring SE equal ($z/2+r_z$,$x/2+r_x$,$y/2+r_y$), where $r_z$, $r_x$, $r_y$ are the spatial shifts randomly picked from the intervals (-$z/4$,$z/4$), (-$x/4$,$x/4$), (-$y/4$,$y/4$). In other words, the neighbouring SE strongly overlap with each other - as can be seen in Figure 6(c).

Looking at the stochastic matrix in Figure 6(a), one can observe that the scale, shape and direction of the perturbations vary from one geological unit to another. For example, perturbations in the prism are finer than those in the crust or mantle. Also their shape in the oceanic crust and prism are more anisotropic than the shape of those implemented in the mantle. In practice, this is implemented by summation of 3D stacks containing different SE. For example, we create stochastic perturbations in the sedimentary layer and the outer prism with only the SE shown in Figure 6(e-f); Perturbations in the inner wedge and underplated units additionally incorporate SE shown in Figure 6(d). Finally, the oceanic crust contains SE at all scales (Figure 6(c-f)). This also applies to the mantle and continental crust, however for these units we use SE with different correlation lengths ratio equal to 0.5 × 1 × 1.

In Figure 6(a), we also show that the stochastic perturbations in some of the units are oriented according to the dipping or bending of the structure. Since we know the polynomials defining each of the interfaces in the model (and therefore their position), we shift vertically the columns of the matrix containing stochastic perturbations of a given geological unit such that they follow the shape of the structure. For example, the perturbations which fill the oceanic crust follow the smooth trend generated from the polynomial approximation of the interfaces creating the top of the oceanic crust. Note, that because we stack truly 3D SE within the 3D cube of the same size as the 3D model, our stochastic perturbations follow continuously the geological structures not only along the inline direction but also along the crossline direction.

To analyse what the designed stochastic perturbations mean in terms of the spatial resolution of the model, we compute the 2D wavenumber spectrum (displayed in logarithmic amplitude scale in Figure 6(h-i)) of the $V_p$ model with and without stochastic





components. As shown in Figure 6(h), the spectral amplitudes of the model without stochastic perturbations are focused around
the low wavenumber part of the spectrum, which represent structures the size of which is of the order of ∼500 m and larger.
In contrast, the high-wavenumber amplitudes are clearly magnified in the spectrum of the model with stochastic components
(Figure 6(i)). The overall distribution of the energy added to the background medium (with respect to the spectrum shown in
Figure 6(h)) is close to normal, which highlights the randomness of the perturbations. The designed approach led to pertur-
bations of smaller size being in fact of the order of the grid size (25 m). On the other hand, one can also observe in Figure
6(i) a few dipping bands of increased amplitudes. They correspond to the anisotropic shape of the SE following the geological
structures - and therefore indicate that our stochastic perturbations are partially predefined and not purely random.

### 2.4.2 Structure warping

Finally, we apply a second type of stochastic components to the model after projection, parametrization and application of
small scale stochastic perturbations. The aim of this step is to perform a small-scale point-wise warping of the input 3D grid
using a distribution of random shifts. By small-scale we mean that the structural changes introduced by warping are in general
much smaller than those incorporated during the projection step. Their impact on the shape of the wavefield will therefore be
weaker than the one induced by the geological structure itself, however more pronounced than that resulting from the small
scale perturbations described in previous section. The idea was inspired by Dynamic Image Warping (DIW) (Hale, 2013). DIW
is a technique which allows for the estimation of local shifts between two images (2D matrices), assuming that one of the image
is a warped version of the other image. This problem is cast as an inverse problem where the unknowns are the shifts that will
allow for matching between the two images.

In our case, we use DIW in a forward sense. This means we build a matrix of smoothly varying random spatial shifts of a
desired scale and magnitude. Then, we warp each of the inline section from the 3D model grid using those shifts. Figure 7(a-b)
shows two 300 km × 300 km matrices that represent distribution of shifts for two different spatial scales. While the scale of
the shifts in Figure 7(a) is local (starting from ∼10 km), the spatial extension of the shifts in Figure 7(b) is regional. The blue
and red colours indicate the negative and positive shifts, respectively - scaled between ±2km. We take the average of the two
distributions to obtain the final distribution shown in Figure 7(c). Our procedure of warping is implemented as follow.

First, for each of the 4001 inline sections, we extract from the distribution shown in Figure 7(c) a 2D sub-matrix with the size
of the inline section. Figure 7(d) shows such a matrix corresponding to the first inline section. It corresponds to the black-
dashed rectangle in the upper-right corner of Figure 7(c). The further rectangles in Figure 7(c) illustrate how we move from
one sub-matrix to another one for five inline sections spaced 20 km apart (inline number 1, 1001, 2001, 3001, 4001).

Once we have the sub-matrix of shifts and the initial inline section $m_i(x,z)$, the warped inline section $m_w(x,z)$ is obtained
by the identity $m_w(x,z) = m_i(x, z+s)$, where $s$ is the entry of the sub-matrix of shifts at the $(x,z)$ position. In Figure 7(d),
the warped structure (dashed black lines) is shifted downward and upward with respect to the initial structure (solid grey line)
for negative (blue colour) and positive (red colour) shifts, respectively. For simplicity, we consider only vertical shifts in this
example. However, we also implement warping in the horizontal direction, i.e. $m_w(x,z)=m_i(x+s,z)$, in the same way as we





do in the vertical direction.

Once an inline section has been warped, we extract a sub-matrix of shifts for the next inline section from the distribution of
shifts shown in Figure 7(c). We perform this extraction at a slightly modified position with respect to the previous one, typically
one grid point further (i.e. 25 m). In this way we maintain the continuity of the warped structures along the crossline direction.
Although this inline by inline procedure is based on the 2D warping, in practice it introduces fully 3D structural changes in the
model, at the same time remaining more practical for quality control of the results and computational efficiency.

Three depth slices extracted at 5 km, 7.5 km and 10 km depth from the 3D $V_p$ models before and after warping highlight how
the warping has modified the shape of the structures (Figure 7(e-g)). Importantly, the continuity of the structure is preserved
during warping not only along the inline direction but also along the crossline direction. It is also worth remembering that, in
this example, the distribution of shifts is random. However, nothing prevent to use a deterministic distribution, for example to
produce uplift or bending of certain parts of the model.

## 3    Wavefield modelling

In this section, we perform full wavefield modelling in a target of the GO_3D_OBS model to assess the footprint of the
structural complexity and the heterogeneity of the physical parameters on the anatomy of these wavefields. Figure 8 shows the
perspective view of the full 3D $V_p$ model with a chair-plot representation. To perform our modelling experiments, we select a
100 km $\times$ 20 km $\times$ 30 km target in the central part of the full model (for each parameter) (white lines in Figure 8). We simulated
OBS gathers, for the station located 10 km apart from the nearest edges of the target (see the white point in Figure 8) and the
air-gun shots at 10 m depth below the sea level. The shooting line is oriented in the inline direction (red lines in Figure 8) at
the position of the inline number 2601. In the next sections, the 2D simulations are performed in this inline model. The source
signature is a 1.5 Hz Ricker wavelet (spectrum energy up to 4.5 Hz) and the propagation time is 30 s. We perform modelling
taking advantage of the reciprocity principle - namely, we place the pressure sources at the position of the hydrophone of the
OBS and extract the pressure component at the position of the air-gun shot. In the following section, we present few examples
of 2D and 3D modelling for different approximations of wavefield propagation. We run the simulations in the time domain
using the TOYXDAC visco-acoustic finite-difference ($2^{nd}$ order in time and $4^{th}$ order in space) and SEM46 visco-elastic
spectral-element FWI codes developed within the framework of the SEISCOPE consortium (https://seiscope2.osug.fr).

### 3.1    Acoustic versus visco-acoustic 2D wavefields

The first modelling test demonstrates the influence of the viscous effects on 2D acoustic P-wave modelling. We compare 2D
wavefields that are computed without ($Q_p$=10000) and with attenuation. The intrinsic attenuation mechanism we use is based
on the generalized Maxwell body including 3 standard linear solid attenuation mechanisms (Yang et al., 2016). For the sake of
computational costs, all the models are re-sampled in a uniform Cartesian grid with a 50 m grid interval.

Figure 9(a-b) shows the OBS gathers generated with constant and variable $Q_p$ models. True-amplitude seismograms are plotted
in both panels with the same amplitude scale. Although the two gathers look almost identical, the difference between them -



shown in Figure 9(c) - clearly shows the variations of the signal amplitudes. Changes are mostly visible in the first arrivals up
       to 15 km offset (corresponding to the waves travelling in the shallow parts of the model), as well as in the energetic reflection
       from the top of the subducting oceanic crust. This reflection is affected by the high attenuation layer within the subduction
       channel.

       Importantly, the variable $Q_p$ model introduced not only differences in terms of wavefield amplitude, but also notable phase
shifts due to dispersion effects. The inset in Figure 9(c) shows the zoom on interleaved traces extracted within the dashed
       rectangles in Figure 9(a-b). Arrivals in the blue-shaded traces (data modelled with $Q_p$=10000) are clearly shifted in time with
       respect to those simulated with variable $Q_p$ model. Therefore, incorporating attenuation model during seismic imaging can not
       only improve the amplitude reconstruction, but it can also have a second order impact on the kinematic correctness of the final
       image.

**3.2   2D versus 3D visco-acoustic wavefields**

       We assess now the footprint of 3D effects by comparing the synthetic seismograms computed by 2D and 3D wave modelling.
       The data resulting from both simulations are shown in Figure 10(a-b). For better readability of far-offset traces, we weight the
       amplitudes by the absolute value of the offset. Both gathers have, of course, different amplitude versus offset variations due
       to the different kinds of sources used in 2D and 3D (line sources versus point sources). Apart from this different amplitude
behaviour, both gathers look similar at the first sight. To better visualize potential 3D kinematic effects, we interleave 20 traces
       from one gather with the following 20 traces from another one and we normalize each trace using RMS value. As a result,
       we obtain the gather presented in Figure 10(c) (the blue-shaded phases correspond to the 2D modelling). Not surprisingly, the
       mismatch between the two sets of seismograms increases in general with increasing offset and time. For instance, the difference
       between the first-arrival traveltimes for the Pn wave is reaching ~150 ms at far offset. However, we also clearly see small time
shifts at short and intermediate offsets and more significant time mismatches for the complex package of reflections between
       20 km and 60 km offset.

       To better understand how the wavefield propagates within the 3D target during the simulation, we extract snapshots every
       2.5 s at 10 km depth (see black dotted line in Figure 8 for the depth-slice location). For comparison, we also perform 3D
       modelling in the 2.5D version of the inline section number 2601. Depth-slices extracted from the $V_p$ 3D and 2.5D models
are presented in Figure 11(a). The corresponding snapshots are displayed in Figure 11(b-l). The lower-half of each panel (the
       blue-shaded phases) presents the respective snapshot from the wavefield modelling performed within 2.5D model. It is clear
       that with increasing time and offset-distance the wavefield simulated in the 3D target becomes more and more complex - as
       compared to the 2.5D counterpart. In particular, while the blue-shaded wavefield is symmetric with respect to the shooting
       profile, the wavefield computed in the 3D target shows evidence of significant off-plane propagation. This is indicated both by
the slant appearance of the first arrival wave (Figure 11(d-f)), as well as the shape and the spatial location of the reflections.
       In Figure 11(m) we show analogous comparison as in Figure 10(c) between 3D and 2.5D seismograms. Although the 2.5D
       data fit better than pure 2D data to their 3D counterparts (probably due to the consistent point-source implementation for 2.5D
       and 3D modelling) the phase shifts between far-offset traces can still be observed. There is also visible mismatch in therms of





amplitude and phase between the traces from the inset containing complex reflections package.

Note, that although our 3D target contains part of the curved subduction front, the acquisition profile is still mainly aligned with the dip direction of the structures within the wedge. Nevertheless, the off-plane wavefield propagation is yet remarkable. One can imagine that this effect can be further magnified when the geological setting contains more heterogeneities along the crossline direction.

### 3.3  Acoustic versus elastic 3D wavefields

The final modelling example which we present here shows the difference between 3D spectral-element acoustic (in the whole model) and acoustic-elastic (acoustic in water and elastic in solid) wavefield propagation. The solid-fluid coupling along the bathymetry is implemented by means of displacement potential (Ross et al., 2009). For the acoustic-elastic test, we extract the target from the full $V_s$ model (Figure 5(a)). The lowest $V_s$ value in the most shallow sediments covering the backstop area of the model reaches ∼530 m/s. To speed up the elastic wavefield simulation we clip $V_s$ to 800 m/s. This makes possible to use

the smallest element size in the spectral-element mesh equal to ∼ 213 m (considering the 1.5 Hz Ricker source and $4^{th}$ order polynomial interpolation). Figure 12(a-b) shows the two OBS gathers inferred from acoustic and elastic wavefield modelling. One can observe that the package of wide-angle reflections between 20 and 60 km in Figure 12(a) contains significantly more energetic arrivals as compared to the same arrivals in Figure 12(b). Some of them are difficult to track (or simply not present) in the gather generated using elastic wavefield propagation. This loss of energy can be explained by the P to S conversions.

In contrast, we see in the elastic seismograms some weak arrivals of low apparent velocity between 10 and 20 km, indicating some elastic effects (Figure 12(b)).

The analogous comparison as in Figure 11(c) is presented in Figure 12(c). One can observe that the waveforms from the acoustics modelling (blue-shaded phases) do not create continuous arrivals with their elastic counterpart delineating differences between two types of data.

It is also worth to mention here, that the data simulated with the finite-difference scheme exhibit some evidence of so called "stair-case effect" (see diffraction-like patterns at the short-offset data in Figure 10(a-b)). These artefacts result from the bathymetry projection on the Cartesian grid (50 m grid step). In contrast, the spectral-element mesh can handle accurately the sharp seabed interface which is reflected by lack of similar artefacts in the gathers from Figure 12.

## 4  Discussion

### 4.1  Geological setting of GO_3D_OBS

Our previous studies oriented on crustal-scale seismic imaging from OBS data were located in the Tokai area of the Nankai Trough (Górszczyk et al., 2017, 2019). This gave us a true-earth reference to guide the building of a first version of our model. However, our primary aim was to build a structural model which involves as many as possible geological features observed in subduction zones rather than following rigorously the Nankai Trough geological setting. On this basis, the GO_3D_OBS





shall be seen as a generic crustal-scale seismic imaging benchmark model rather than a seismological reference to the Tokai segment. We hope that, based on some feedback from the community, we will be able to incorporate more geological features or modify existing ones.

## 4.2   Acquisition design

In the academic community of marine geosciences, sparse OBS acquisitions remain the primary tool for offshore lithospheric
imaging. The acquisition-related issues focus mainly on the shooting strategy and the optimal OBS spacing (Brenders and Pratt, 2007a). This is the consequence of the fact that the number of available instruments is limited ( $\sim 100$ maximum). The spacing between OBS must not be chosen at the expense of the maximum offset sampled by the acquisition as this maximum offset should be enough to record diving waves that reach the deepest structures (namely the upper mantle). On the other hand, the sparse acquisition design shall minimise model distortions resulting from an incomplete structure illumination associated
with limited coverage of the surface acquisition. These distortions need to be assessed since they can generate significant bias during geological interpretation.

To investigate the footprint of various acquisition setting on the imaged target, realistic synthetic models are necessary for the survey design. Processing of different datasets generated in such models would allow for optimization of the acquisition setup to find the best compromise between deployment effort, image resolution and target sampling. This was our main motivation
beyond the design of the GO_3D_OBS model. One of the ongoing projects now focuses on the investigations of the off-plane wavefield propagation during 2D crustal-scale FWI. From the modelling tests shown in this paper (Figure 10), we can conclude that these effects may have a detrimental impact on the imaging results. This shall stimulate the extension of routinely performed 2D OBS surveys toward the optimally designed future 3D deployments. We can evoke here the 2001 Seize France Japan (SFJ) 2D OBS experiment (Operto et al., 2006) which was conducted in the Tokai area of the Nankai Trough using
100 OBS deployed with a dense 1 km spacing. Today, 3D sparse OBS deployments are conducted with a similar number of stations with an aim to apply FWI. Therefore which of those two different acquisition setups are closer to the optimal setting? Was the 2D SFJ profile oversampled or maybe the today's 3D experiments are undersampled? We believe that for a new OBS experiments we shall look for justification of a proposed acquisition geometry rather than use an ad hoc configuration.

## 4.3   Usefulness for tomography

Although the processing method we pair the most with our model is FWI, there is no limits in terms of application of other tomographic techniques, for example ray-theory based modelling techniques such as ray-tracing or eikonal solvers. While FAT is typically producing rather smooth velocity models, the other variants of tomography have potential to build models with improved resolution. Evaluation of those tomographic methods against GO_3D_OBS model can lead to their further development. Traveltime tomography techniques are in general computationally much less intensive than waveform inversion
methods. It is therefore worth to push their efficiency to the resolution-limits and validate them with complex benchmarks. This kind of development of tomographic methods indirectly contributes also to the FWI popularisation since the tomographic models are usually used as initial models for FWI.



### 4.4 Uncertainty estimation

Together with development of different imaging techniques, methods for quantitative estimation of their uncertainty and res-
olution limits shall be proposed. Indeed, this subject is not trivial when facing real-data processing and therefore the issue is
often skipped by authors. The obvious problem is that the true geological structure is never exactly known, and therefore the
model of this structure - derived under given approximation - has no direct reference point to compare. On the other hand,
the resolution and accuracy of the seismic imaging methods are always limited. Therefore, it shall bring us to reflection about
the potential pitfalls related to over-interpretations during real-data case studies. Evaluating the uncertainty of the inversion
methods with realistic synthetic tests can bring us closer to the estimation of the error during real-data processing. Not only
because we exactly know the underlying structure, but also because of the possibility to investigate the influence of different
approximations, experiment geometries or noise.

### 4.5 Further development

At the current stage of development, we believe that GO_3D_OBS is sufficiently realistic to explore the validity of various
tomographic and inversion methods on synthetic datasets with different acquisition designs. Future branches of development
of this benchmark can cover different aspects including both structural components and parametrisation. For instance, one of
the interesting topics is extension toward anisotropy. Building such an anisotropy model consistent with geological studies of
deep crust could be challenging due to general difficulty of precise anisotropy estimation. Nevertheless, it would significantly
increase authenticity of the corresponding seismic data. On the other hand, generating accompanying dataset for the current
models can further broaden its impact. Possible dataset could include sparse 3D and dense 2D OBS deployments, as well as
corresponding streamer data. Such dataset could be directly use for a given type of processing. In particular, generating the
seismic data without and with realistic type and level of noise would help to further evaluate the impact of noise on the accuracy
of the seismic imaging methods.

## 5 Conclusions

We developed the GO_3D_OBS reference geomodel for the purpose of assessing different crustal-scale seismic tomographic
and inversion methods, as well as for guiding various seismic acquisition designs. The 3D model structure allows for ex-
traction of sub-volumes representing different level of complexity which can be found in subduction zone environments. The
visco-elastic parametrization of the medium gives the potential user the possibility to assess the impact of various physical
approximation during imaging. There are various components influencing the current state-of-the-art in terms of crustal-scale
imaging. Among them, methodological developments aiming at computationally-efficient seismic wave modelling and robust
inversion schemes assessed against realistic large-scale problems significantly contribute to the further exciting geological dis-
coveries. In this perspective we believe that our model will help to further stimulate high-resolution seismic imaging of deep
targets and through this will help to understand the processes shaping the Earth's crust.



*Data availability.* The current version of model and the user-manual are available from the archive on https://geoimaging.igf.edu.pl/go_3d_obs.

*Author contributions.* AG and SO initiated the research and set-up the overall concept of the 3D model. AG designed and implemented the workflow for building the model structure, the physical parametrization and the stochastic components. SO validated each of the model ingredients. AG designed and performed seismic modeling simulations. AG prepared the initial version of the manuscript which was further updated toward the final version with the contribution from SO.

*Competing interests.* The authors declare no competing interests

*Acknowledgements.* This study was partially funded: (i) by the SEISCOPE consortium (http://seiscope2.osug.fr), sponsored by AKERBP, CGG, CHEVRON, EQUINOR, EXXON-MOBIL, JGI, PETROBRAS, SCHLUMBERGER, SHELL, SINOPEC, SISPROBE, and TOTAL; (ii) the Polish National Science Center, (grant no: 2019/33/ B/ST10/01014). The study was granted access to the HPC PL-Grid Infrastructure (grant id: 3dwind). We thank Romain Brossier and Jian Cao for their support of this study through the development of TOYXDAC and SEM46 codes, as well as Ludovic Metivier and Jean Virieux for the internal review of the manuscript. We thank geological community from
Geoazur and JAMSTEC institutes for their feedback which allowed us to improve the structure of our model.





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



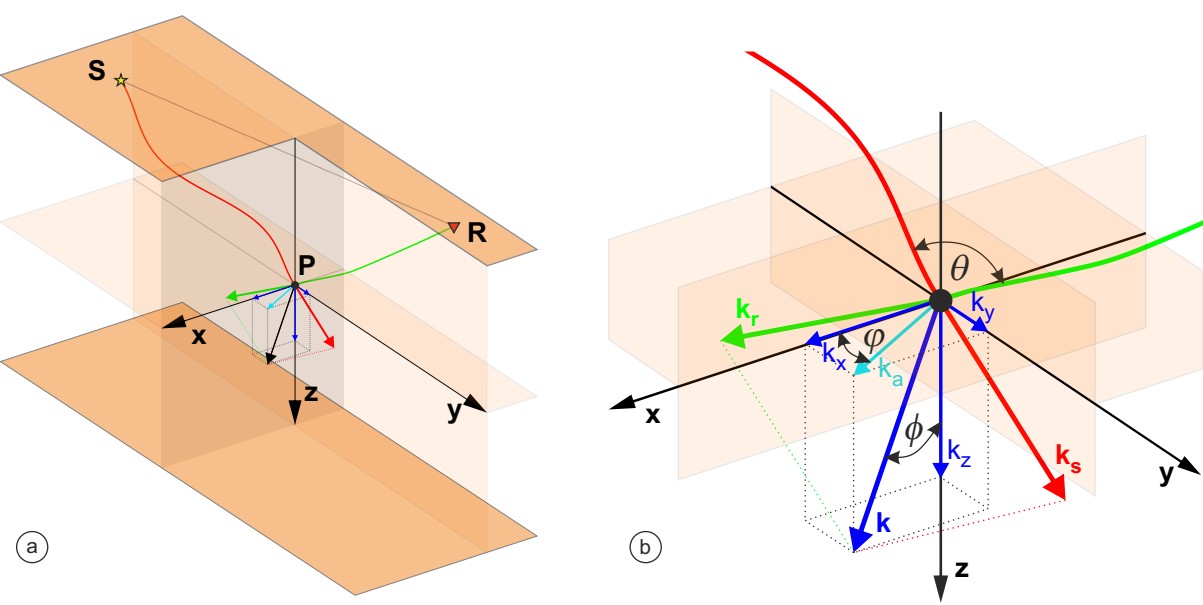

**Figure 1.** (a) Sketch of the single source/receiver (**S/R**) acquisition and the 3D wavenumber vector **k** mapped by FWI at the scattering point **P**; (b) Zoom on (a). Local wavenumber vector **k** is a sum of **k$_s$** and **k$_r$** vectors associated with the raypaths emerging from the source **S** and receiver **R** creating the scattering angle $\theta$ at the scattering point **P**. Dip and azimuth of the wavenumber vector **k** is defined by the $\phi$ and $\varphi$ angles. The modulus of **k** is given by $(\lambda/2)\cos(\theta/2)$ where $\lambda$ denotes the local wavelength (Miller et al., 1987; Wu and Toksöz, 1987). The range of **k** mapped at each point **P** by the acquisition gives the resolution with which the subsurface is reconstructed by FWI.

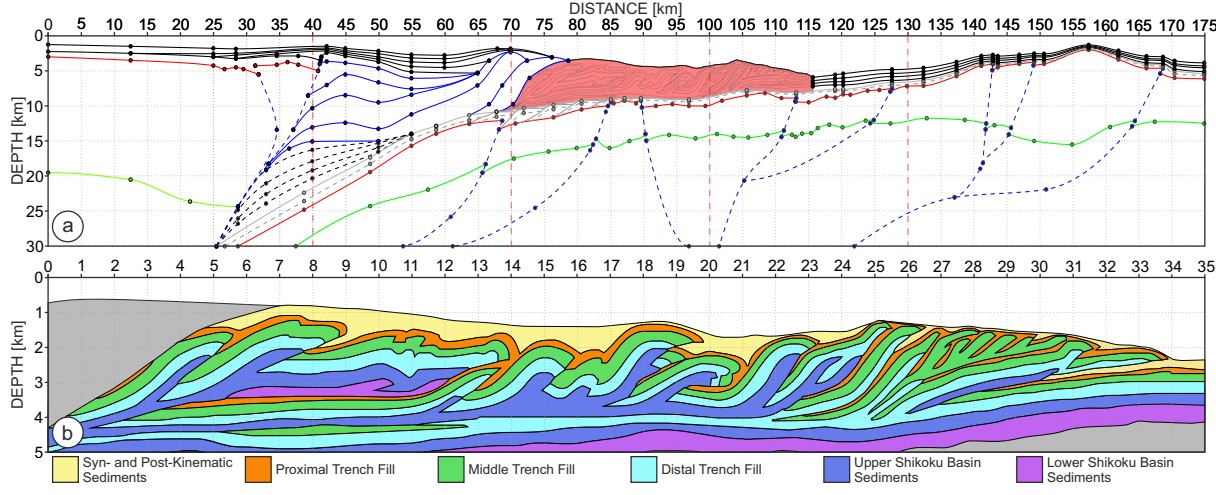

**Figure 2.** (a) 2D structural skeleton - inline section at y=0 km; (b) Structure of the accretionary prism adopted form (Kington, 2012).





**Figure 3.** (a) Geometry of the major fault-planes in the model (nodes at every 5 km are marked); (b) Projection of the initial structure from Figure 2(a) into 3D (sections at every 20 km are extracted); (c-f) Crossline sections extracted at 40 km, 70 km, 100 km and 130 km - vertical red dashed lines in Figure 2(a).



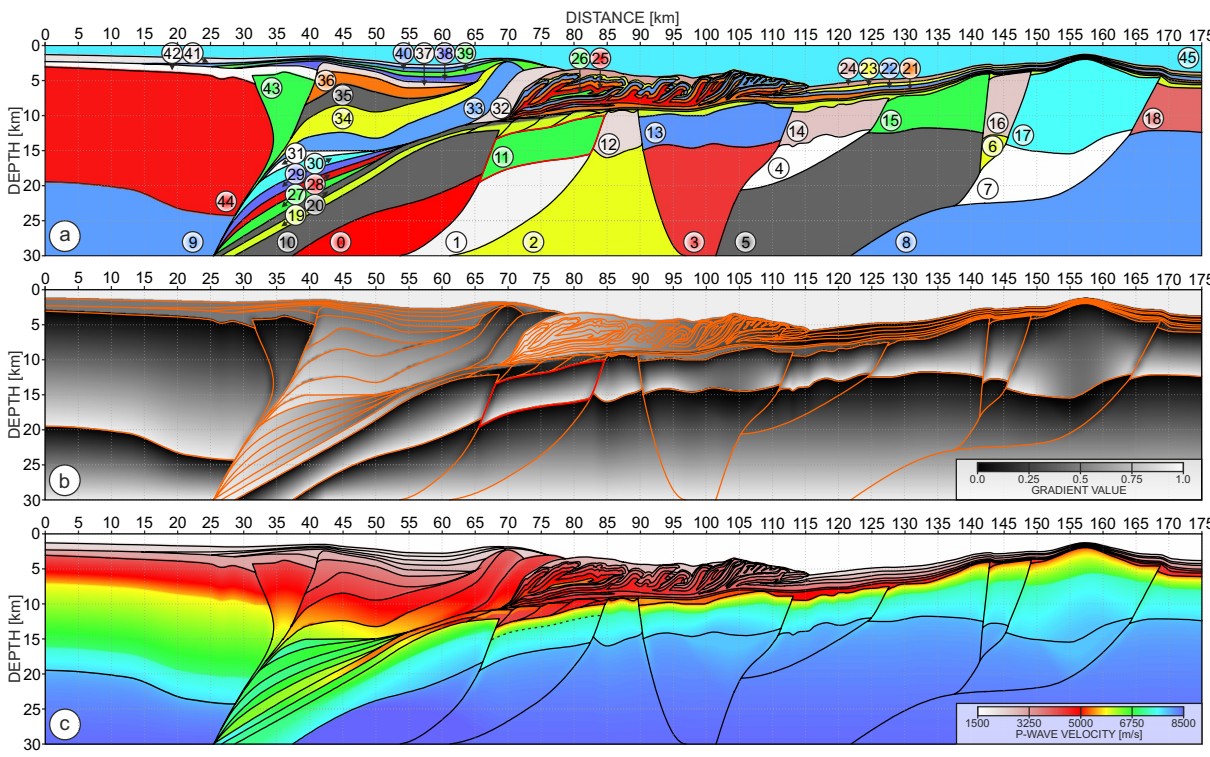

**Figure 4.** (a) Index matrix with the 46 geological units marked by randomly picked colours. (b) Gradient matrix used for implementation of spatial variations within those units. (c) 2D inline obtained after assigning $V_p$ values to the matrices shown in (a) and (b).





**Figure 5.** (a) $V_s$, (b) $\rho$, (c) $V_p/V_s$, (d) $Q_p$, (e) $Q_s$, (f) $Q_p/Q_s$ models derived from the $V_p$ inline from Figure 4(c).



**Figure 6.** (a) Matrix of stochastic perturbations for a 2D inline section. (b) $V_p$ model shown in Figure 4(c) after application of the stochastic perturbations shown in (a). (c-f) 3D stack of structural elements of different scales. Red/blue colours indicate positive/negative magnitudes of the structural elements. (g) Shape of the 3D structural element used to generate the stochastic perturbations. (h-i) Wavenumber spectrum of the 2D $V_p$ inline section with and without stochastic perturbations.



**Figure 7.** (a)-(b) Spaces of random small- and large-scale spatial shifts respectively; (c) Final space of shifts - average from (a) and (b). Dashed rectangles mark the matrices of shifts for in-lines at each 20 km of crossline distance; (d) Matrix of shifts extracted from (c) (black dashed rectangle number 1). Gray solid and black dashed lines represent the structure before and after warping respectively; (e)-(g) and (h)-(j) Depth-slice sections extracted at 5 km, 7.5 km, 10 km before and after warping respectively.

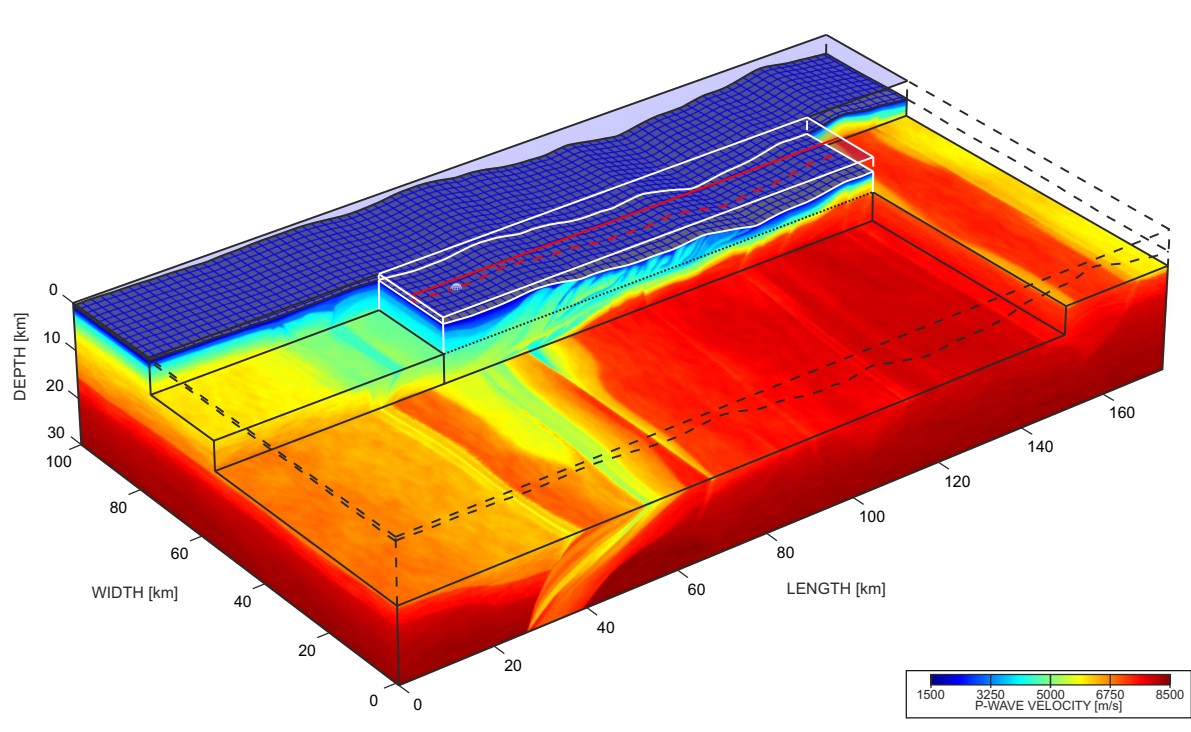

**Figure 8.** The perspective view on the chair-plot of the 3D $V_p$ cube. The central volume defined by the white lines corresponds to the part of the model (100 km × 20 km × 30 km) used for simulation of the data presented in Section 3. The white point at the seabed marks the OBS position and the red solid and dashed lines denotes the shooting profile and its projection to the bathymetry respectively.



**Figure 9.** OBS gathers generated using 2D finite difference (a) acoustic and (b) visco-acoustic waveform modelling. (c) Difference between
(a) and (b). The inset shows time shifts between interleaved traces from (a) (blue shading) and (b) extracted within green dashed rectangles.

**Figure 10.** OBS gathers generated using finite difference visco-acoustic (a) 2D and (b) 3D waveform modelling. (c) Comparison of seismograms from (a) and (b) where 20 traces from 3D modelling are interleaved with the following 20 traces from the 2D equivalent (light-blue shading). For better readability, the seismograms in (a) and (b) are scaled with the absolute offset value, while the traces in (c) are divided by their RMS.



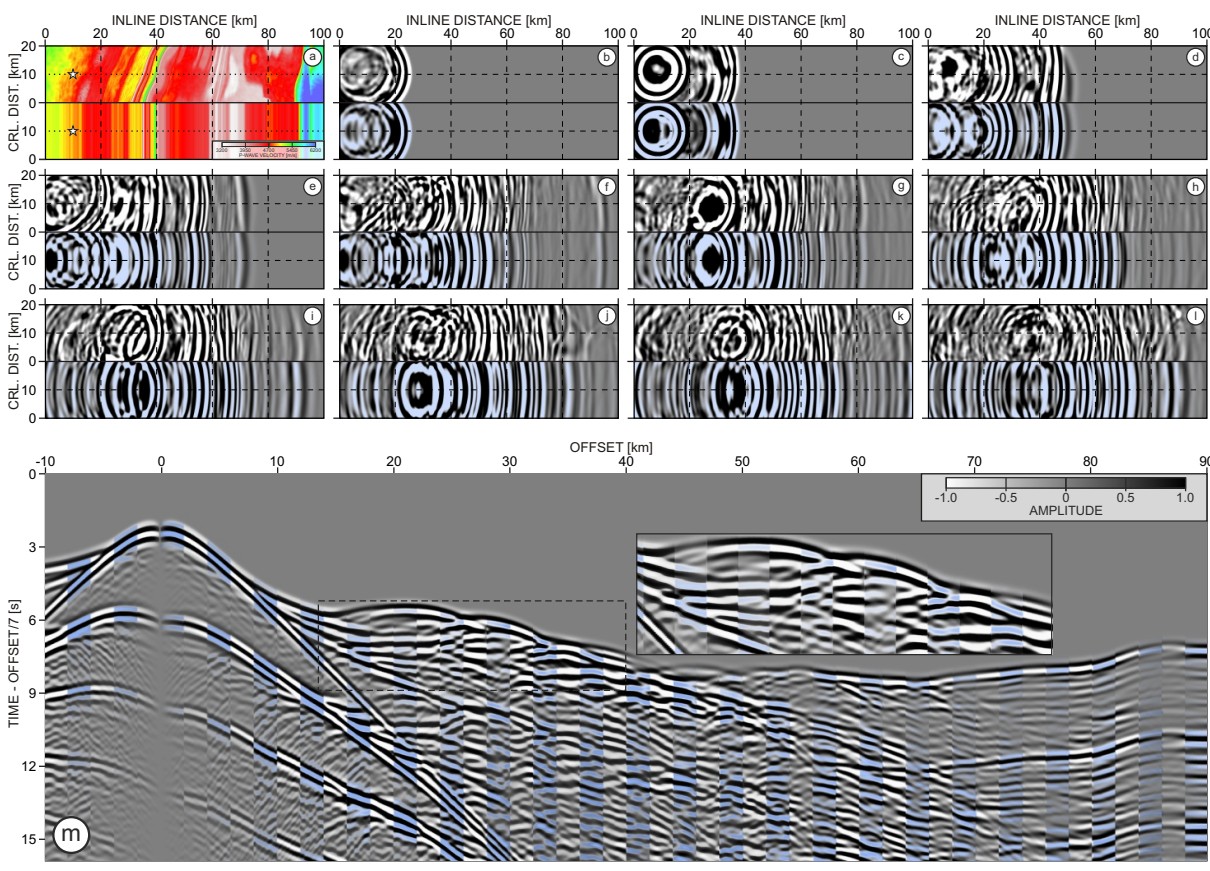

**Figure 11.** (a) Depth-slice sections extracted at 10 km depth from 3D (upper half) and 2.5D (lower half) $V_p$ sub-volumes. White stars and black dotted lines mark the projection of the OBS position and shooting profile respectively; (b)-(l) Snapshots extracted during wavefield modelling at each 2.5 s time-step along the corresponding depth-slices from (a). (m) Analogous comparison between 2.5D and 3D data as in Figure 10(c). Amplitudes are scaled with the absolute offset value.



**Figure 12.** OBS gathers generated using 3D spectral-element (a) acoustic and (b) elastic waveform modelling. (c) Comparison of seismograms from (a) and (b) where 20 traces from acoustic modelling (light-blue shading) are interleaved with the following 20 traces from the elastic equivalent. For better readability, the seismograms in (a) and (b) are scaled with the absolute offset value, while the traces in (c) are divided by their RMS.