# Peer review of "GO\_3D\_OBS - The multi-parameter benchmark geomodel for seismic imaging methods assessment and next generation 3D surveys design (version 1.0)"

_Geoscientific Model Development, 2020_

## Short Comment (SC1) · 7 Oct 2020

No comment on the proposed benchmark except it looks beautiful. I just noticed a typo in figure 4 to 7, on top of the figures, DISTANCE is spelled DISTNACE best regards

---

## Author Comment (AC1) · 9 Oct 2020

Thank you for your comment and the keen eye Gautier. I will correct the labels accordingly.

---

## Short Comment (SC2) · 14 Nov 2020

Dear authors,

in my role as Executive editor of GMD, I would like to bring to your attention our Editorial version 1.2:

https://www.geosci-model-dev.net/12/2215/2019/

This highlights some requirements of papers published in GMD, which is also available

on the GMD website in the 'Manuscript Types' section: http://www.geoscientific-model-development.net/submission/manuscript_types.html

In particular, please note that for your paper, the following requirement has not been met in the Discussions paper:

- Code must be published on a persistent public archive with a unique identifier for the exact model version described in the paper or uploaded to the supplement, unless this is impossible for reasons beyond the control of authors. All papers must include a section, at the end of the paper, entitled "Code availability". Here, either instructions for obtaining the code, or the reasons why the code is not available should be clearly stated. It is preferred for the code to be uploaded as a supplement or to be made available at a data repository with an associated DOI (digital object identifier) for the exact model version described in the paper. Alternatively, for established models, there may be an existing means of accessing the code through a particular system. In this case, there must exist a means of permanently accessing the precise model version described in the paper. In some cases, authors may prefer to put models on their own website, or to act as a point of contact for obtaining the code. Given the impermanence of websites and email addresses, this is not encouraged, and authors should consider improving the availability with a more permanent arrangement. Making code available through personal websites or via email contact to the authors is not sufficient. After the paper is accepted the model archive should be updated to include a link to the GMD paper.

As a simple link to a website is not at all a persistent access, before final publication, a persistent archive for the exact code version discussed in this paper needs to be provided.

Yours, Astrid Kerkweg

---

## Author Comment (AC2) · 15 Nov 2020

Dear Astrid,

Thank you for highlighting the GMD requirements regarding the data and code availability. I would like to underline that a long time before the submission we encountered problems finding a pubic data-hosting service that could store our files which are in overall more than 1 TB.

Recently we managed to store all available model-related data in the public datarepository of Institute of Geophysics, Polish Academy of Sciences:

https://dataportal.igf.edu.pl/dataset/go_3d_obs

which warrants the persistent and open access to the model. DOI number will also be added once the manuscript will be published.

Of course we will edit accordingly the manuscript sections related to the data and code availability before final publication.

Best regards, Andrzej Górszczyk

---

## Referee Comment (RC1) · Anonymous Referee #1 · 27 Nov 2020

The manuscript "GO_3D_OBS - The Nankai Trough-inspired benchmark geomodel for seismic imaging methods assessment and next generation 3D surveys design (version 1.0)" by Górszczyk and Operto is an excellent, useful and timely contribution to the field of geophysics and, in particular, for seismic imaging and inversion. The overall goal is the design and building of a detailed 3D benchmark geophysical model with a visco-elastic parameterization that represents a subduction zone. I am actually impressed by the level of detail and meticulousness put in all the steps of the work. It integrates a large number of consistently designed and conformally shaped bodies,

layers and structures that represent highly realistic geological and tectonic features present at many subduction zones. Despite the large degree of structural complexity, the model is constructed following a well-designed and logically structured sequence of steps so that the final result preserves an astonishing level of "geological realism". The steps of the process are designed to sequentially incorporate an increasing level of complexity and details to the model but, at the same time, they are isolated and flexible enough to change any of the attributes of the model or to construct a different one if it were necessary. My main concern is that, while detailed, the technical information provided in the manuscript is not sufficient for a motivated reader to reproduce the model itself. The description of the relationships and equations applied to perform the transformations at each step is only general and not specific for the different units. The editorial team should consider whether this is acceptable or the issue should be addressed (probably by adding a large volume of supplementary material). In any case, I would like to thank and congratulate the authors for the thorough and rigorous work, which I find particularly useful for the years to come. In summary, I consider that it deserves to be published after a minor and limited revision only. I have a number of minor comments and observations made while reading the manuscript, although I note that a few of them are partially addressed in other sections of the manuscript.

Minor comments:

Title: The reference to Nankai trough is perhaps too specific for a title. Consider changing it for a general reference to a subduction setting, it could be more effective to attract wider readership attention. The specific reference to Nankai can be included in the abstract and text.

Line 9: Did you ever consider adding anisotropy? Why did you decide not to?

Line 24: I would add joint refraction and reflection travel-time tomography in either 2D (Korenaga et al., 2000) or in 3D (Melendez et al., 2015)

Korenaga, J., et al, 2000. Crustal structure of the southeast Greenland margin from

joint refraction and reflection seismic tomography, J. geophys. Res., 105, 21 591–21 614.

Meléndez, A., et al., TOMO3D: 3-D joint refraction and reflection traveltime tomography parallel code for active-source seismic data – synthetic test, Geophys. J. Int., 203, 158–174, 2015

Line 110: Exploration of aternative options for robust objective functions also in Jimenez-Tejero et al (2018)

Jimenez-Tejero, C., et al. Appraisal of Instantaneous Phase-Based Functions in Adjoint Waveform Inversion, IEEE Transactions on Geoscience and Remote Sensing, 56, 9, 5185 - 5197, 2018 Line 134: "of subduction zones"

Line 135: "As an experimental"

Line 144-145: "The empirical components impose the physical parametrisation (...) in terms of the magnitude of subsequent parameters and relations between them" > It is unclear to me what you mean here. Could you rephrase or clarify it further?

Line 147: What do you mean by "realistic sructure variations"?

Line 154: "as follows"

Line 163: "features which were interpreted" > either "features that were interpreted" or simply "features interprteted" ("which" goes after a comma)

Line 166: "was designed"

Line 184: " It currently approaches the subduction zone and simultaneously undergoes the thrusting process" > How can it do both thinkgs at the same time? Approaching prior to thrusting is closer to what is shown

Line 228: Equation 1 would require some extra explanation. As it expressed, the right hand sides give always the same value. I mean, explain a bit how do coefficients

a,b,c,d,e vary in different domains

Line 241: "of another"

Line 293 and Fig. 4b: What are the criteria to set the gradient matrix values in the different units? Most have vertical gradients reflecting compaction or lithological burden but others do not seem to. Please elaborate a bit on this

Line 294: Why is there no -vertical- gradient in the shallow sediments? It is typically a place where changes of properties with depth by compaction are strongest

Equation 3: G1=Gn*3.3 (remove parentheses)

Line 315: Vp=7.8 km/s is far too high for oceanic L3

Line 326: "subducting volcanic ridges" or "subducting seamounts"

Line 339: What do you mean by "second-order parameters"? Which are the "first-order" ones?

Line 351: Brocher's (2005) is an empirical relationship with significant uncertainty/error bounds. Applying the exact same relationship (same polynomial conversion law) in all units and sectors sounds like too "perfect". In particular this approach "fixes" Vp/Vs. Wouldn't had been better to define and apply slighlty modified versions of the conversion laws in different units/sectors, too? Would this have any effect at all on FWI?

Line 352: "on laboratory measurements, (...)"

Line 366: thermal effects, too?

Line 371: Same as in Vs(Vp): wouldn't it be better to allow for a range of variation in the shape of conversion laws to allow for heterogeneous Qp/Qs?

Line 390: Small-scale perturbations: Interesting approach, it makes all sense introducing small-scale perturbations, although the selection of the size and shape of the SEs seems rather arbitrary in some cases. On the other hand, couldn't it happen that in

some instances the values of the stacked SEs align so that the size of the magnitude is larger than 1? I mean up to 4 or -4 if you are stacking four SEs? Or you re-normalize between (-1,1) after stack?

Line 427: "The overall distribution of the energy added to the background medium (...) is close to normal" > It would be nice to illustrate this with a figure.

Line 443: How are warping matrices in figs 7a and 7b build?

General comment: I wonder whether it is necessary that you give precise information on the expressions used at each step of the process (projection, gradient, physical parameters, stochastic perturbations, warping) applied at each unit, etc. I mean, if you do not do this, your results (in this case, the model) are not fully reproducible. At least not with the information provided.

Figure 9: The figure is excellent, although I do not think that it is the best way to show the effect of variable attenuation. It would probably help showing a few individual traces (seismograms) showing details of the effect for different sectors of the model, offsets and recording times. It would help visualizing not only amplitude but also phase differences.

Figure 10: Same as with fig 9. Showing a few well-chosen traces (additionally or alternatively to the whole records in this fig) could also help visualizing differences and effects. Comment also valid for fig 11m where differences are even slighter.

Line 530: "modelling example, which (...)" or "modelling example that (...)"

Figure 12: Same comment as in the previous two figures concerning comparison of several individual traces.

Line 577-578: An extra question to be considered: what can be gained from joint inversion of spatially coincident OBS and MCS data? Would it somehow mitigate the need of "densely sampled" OBS acquisitions?

Line 590: Uncertainty estimation: While it is true that uncertainty analysis has commonly been overlooked, it is becoming more and more common in recent times. Several schemes have been proposed and it is now routinely done in many travel-time tomography studies. As an example, the description of a formal Monte Carlo sampling scheme-based analysis can ne found in Korenaga & Sager (2012). I'd say that the actual situation deserves a reference in this section.

Korenaga, J. & Sager, W.W., 2012. Seismic tomography of Shatsky Rise by adaptive importance sampling, J. geophys. Res., 117, B08102, doi:10.1029/2012JB009248.

---

## Referee Comment (RC2) · Rie Nakata (Referee) · 19 Jan 2021

The manuscript describes about novel model development efforts based on the authors' previous works in the eastern Nankai Trough. The paper is well written and mostly easy to follow. The developed model will be useful for the community including the model developing methodologies.

I have comments as below. I found two components are missing: i) comparisons of waveforms to observed ones : are the waveforms representative enough? , ii) compar-
isons with drilling efforts or onshore proxy sites for physical properties. Some discussions will be useful.

i) FWI intended

It is fine to build a model with a focus on multiparameter FWI – but slightly differs from title (imaging). You start FWI as "velocity building" which is typically different from "imaging". You may want to clarify these points and perhaps add "FWI" in abstract. Can the authors discuss if we can simulate 3D reflection dataset (as in Kumano) using the model and test various imaging methods too? Are the grid sizes etc sufficient?

ii) IODP drilling efforts and lab experiments, other subduction zones

There are numerous drilling efforts in the subduction zones and field sampling in proxy sites, including those off the Kumano-nada region of the Nankai trough. The physical properties (and so on) should reflect the results. Please add comments on how your model leverage these efforts.

Eastern Nankai?: Kingston's work is off Kumano and thus not eastern Nankai. As your structural model significantly depends on his work, I suggest to remove specific reference to "eastern" and add references of Kumano too.

The authors describe very lightly about applicability to other subduction zone studies. How much does the model applicable and what sense? Is the model applicable to erosional margins? Or the procedure used to build a model? Adding more references is also important.

iii) are there specific problems encountered in pervious FWI/imaging works apart of scaling issues? For example, as seen in Park et al (2010) and addressed in Kamei et al. (2012), low velocity zones were problematic for MVA and subsequent earthquake fault imaging was really a problem (perhaps blank accretionary prisms) – the authors rightly mention "trapped" waves etc. Adding imaging/inversion issues for (specific) important geological features will be helpful (and if they are incorporated into).

"deep" targets: what do you mean by deep targets? Perhaps you can be more specific?

iv) waveform modeling Great to see a range of different modeling efforts. Some motivational statements will be helpful: Why do these modeling important? Are you recommending to generate 3D (visco-)elastic spectral-element waveforms and apply a method of imaging/inversion? Are you going to make these waveforms available as "datasets" as done by BP/Chevron etc?

Interwoven OBS gathers are nice, but some of the authors descriptions are difficult to follow (esp. 2D vs 2.5D vs 3D) unless scrutinizing those plots. The authors should add arrows (e.g. Pn waves or representative off-planer waves). Also it would be easier to understand "complexity" if the authors show 2.5D snapshots along with 3D snapshots. Please add amplitude spectrum and amplitude-vs-offset curves to show the spectra esp. for pure vs visco acoustic simulations to quantitively display the discrepancies.

Please add comments on whether/how much the modeled waveforms represent the observed waveforms (e.g. in eastern Nankai) to convince the model is representative.

A word for choosing spectral element at the start of section 3.3 will be beneficial rather than at the end.

v) imaging The authors discuss about benefits to FWI/tomography/acquisition. How does the model help imaging? How the model help bridging imaging and tomography gaps?

vi) small scale perturbations "disk-shaped structural elements": what do these who geologically? What do they need to overlap?

Additional comments: Figure 1: Is the figure necessary? I do not know if any outside FWI community understands the figure without further expanding the descriptions. The manuscript is about model not FWI. I think the figure is unnecessary.

Figure 2: Add meanings of the lines in Figure 2a in the caption.

[Figure]

Figure 5: Perhaps add a vertical profile?

Figure 6: c-f: what are numbers on the top left? "Red/blue colours indicate. . ." is difficult to follow.

RMS: is not defined.

---

## Author Comment (AC3) · 25 Jan 2021

The manuscript "GO_3D_OBS - The Nankai Trough-inspired benchmark geomodel for seismic imaging methods assessment and next generation 3D surveys design (version 1.0)" by Górszczyk and Operto is an excellent, useful and timely contribution to the field of geophysics and, in particular, for seismic imaging and inversion. The

overall goal is the design and building of a detailed 3D benchmark geophysical model with a visco-elastic parameterization that represents a subduction zone. I am actually impressed by the level of detail and meticulousness put in all the steps of the work. It integrates a large number of consistently designed and conformally shaped bodies, layers and structures that represent highly realistic geological and tectonic features present at many subduction zones. Despite the large degree of structural complexity, the model is constructed following a well-designed and logically structured sequence of steps so that the final result preserves an astonishing level of "geological realism". The steps of the process are designed to sequentially incorporate an increasing level of complexity and details to the model but, at the same time, they are isolated and flexible enough to change any of the attributes of the model or to construct a different one if it were necessary. My main concern is that, while detailed, the technical information provided in the manuscript is not sufficient for a motivated reader to reproduce the model itself. The description of the relationships and equations applied to perform the transformations at each step is only general and not specific for the different units. The editorial team should consider whether this is acceptable or the issue should be addressed (probably by adding a large volume of supplementary material). In any case, I would like to thank and congratulate the authors for the thorough and rigorous work, which I find particularly useful for the years to come. In summary, I consider that it deserves to be published after a minor and limited revision only. I have a number of minor comments and observations made while reading the manuscript, although I note that a few of them are partially addressed in other sections of the manuscript.

**Dear Referee,**

**Thank you for your positive assessment of our work and for your constructive comments. Please find hereafter our answers.**

[Figure]

**Best regards**
**The authors**

Minor comments:

Title: The reference to Nankai trough is perhaps too specific for a title. Consider changing it for a general reference to a subduction setting, it could be more effective to attract wider readership attention. The specific reference to Nankai can be included in the abstract and text.

**We thank you for this comment. We agree that referring to Nankai Trough in the title is too specific according to the objective of the paper which is to propose a crustal benchmark representative of a subduction setting. We change the "GO_3D_OBS - The Nankai Trough-inspired benchmark geomodel..." to "GO_3D_OBS - The multiparameter benchmark geomodel..."**

Line 9: Did you ever consider adding anisotropy? Why did you decide not to?

**We refer to this issue in Section 4.5. At the current stage of development, we wanted to release the isotropic version of the model to reach as large a community as possible before considering more complex parametrization. Extension towards anisotropy is certainly one of the main directions of development. However, it will require an in-depth geological analysis to assign the most suitable anisotropy in terms of symmetry class and strength to each structural units. Also, anisotropy is not well documented in the deep crust.**

Line 24: I would add joint refraction and reflection travel-time tomography in either 2D (Korenaga et al., 2000) or in 3D (Melendez et al., 2015)

*Korenaga, J., et al, 2000. Crustal structure of the southeast Greenland margin from joint refraction and reflection seismic tomography, J. geophys. Res., 105, 21 591–21 614.*

*Meléndez, A., et al., TOMO3D: 3-D joint refraction and reflection traveltime tomography parallel code for active-source seismic data – synthetic test, Geophys. J. Int., 203, 158–174, 2015*

> **Thank you for this suggestion. We add these references related to refraction-reflection tomography.**

Line 110: Exploration of alternative options for robust objective functions also in Jimenez-Tejero et al (2018)

*Jimenez-Tejero, C., et al. Appraisal of Instantaneous Phase-Based Functions in Adjoint Waveform Inversion, IEEE Transactions on Geoscience and Remote Sensing, 56, 9, 5185 - 5197, 2018*

> **We appreciate this suggestion. However, in this place, we refer to the design of more robust types of distances in FWI rather than application of the $L^2$ norm to some specific attribute of the signal - in this case instantaneous phase.**

Line 134: "of subduction zones"

**We correct the sentence.**

Line 135: "As an experimental"

**We correct the sentence.**

Line 144-145: "The empirical components impose the physical parametrisation (...) in terms of the magnitude of subsequent parameters and relations between them" > It is unclear to me what you mean here. Could you rephrase or clarify it further?

**We correct the sentence. We wanted to say that the physical properties in the units are defined according to the results of previous studies and empirical relations between the different parameter classes.**

Line 147: What do you mean by "realistic structure variations"?

**We correct the sentence. We mean realistic variations of the geological structure. We also check that these variations induce significant 3D effects in the wavefields such that the detrimental effects of the 2D assumption in seismic imaging can be assessed more accurately (Figure 10).**

Line 154: "as follows"

**We correct the sentence.**

Line 163: "features which were interpreted" > either "features that were interpreted" or simply "features interpreted" ("which" goes after a comma)

**We correct the sentence.**

Line 166: "was designed"

**We correct the sentence.**

Line 184: " It currently approaches the subduction zone and simultaneously undergoes the thrusting process" > How can it do both things at the same time? Approaching prior to thrusting is closer to what is shown

**It might be true that the thrusting process of Zenisu ridge began once the ridge approached closer to the accretionary wedge. We wont argue about this detail, however in Mazzotti et al. (1999) it is proposed that Zenisu ridge is a compressive structure originated from the N–S shortening of the volcanic Izu-Bonin arc resulting from the kinematic discontinuity along the border of the arc with the Shikoku basin. It is therefore related to the collision of the the Izu arc with central Japan. On the other hand Chamot-Rooke and Le Pichon (1989) proposed, that after the breaking of the crust along Zenisu ridge the subduction of the Philippine Sea Plate proceeds and while the break-point moves towards the trench the thrust is tilting.**

Line 228: Equation 1 would require some extra explanation. As it expressed, the right hand sides give always the same value. I mean, explain a bit how do coefficients a,b,c,d,e vary in different domains

**We reformulate Equation 1 to make it more strict. We also add subscripts "n" to the coefficients $a, b, c, d, e$ to underline that their definition depends on the node "n". Those coefficients are set up and tuned separately for each node in Figure 2a such that, after the projection, they follow the shape of the pre-designed geological structures while guaranteeing the conformity of these structures. In the next paragraph, we discuss possible dependencies between the functions used to project the nodes belonging to a given interface. An illustration of the projection functions that build the main faults in the model is also presented in Figure 3a.**

Line 241: "of another"

**We correct the sentence.**

Line 293 and Fig. 4b: What are the criteria to set the gradient matrix values in the different units? Most have vertical gradients reflecting compaction or lithological burden but others do not seem to. Please elaborate a bit on this

**We discuss this issue earlier in the text. "The spatial variation of the parameters within the same unit can be related to increasing depth in the mantle, layering of the crust, low-velocity zones in the subducting**

[Figure]

**sediments, compaction in the prism or damage zones around the faults etc.". For example, to implement the LVZ within the layers of subducting sediments (subduction channel) - that extend over the whole distance of the model - we need to introduce gradients, which mimic horizontal rather vertical parameter variations. Similar comment applies to the tilted layers representing underplated material where the implemented gradients follow the slope of these structures.**

Line 294: Why is there no -vertical- gradient in the shallow sediments? It is typically a place where changes of properties with depth by compaction are strongest

**The shallow sedimentary layers are relatively thin compared to the other large scale units. Therefore, even for the final model grid size (that is 25 m), a single sedimentary layer is defined by just a few grid points to implement the smooth velocity variation inside this layer. However, the compaction effect in the sediments is introduced by defining the gradually increasing velocities inside the successive layers. There are six thin sedimentary layers in the trench and five in the forearc basin, which lead to quite realistic velocity increase with depth after application of the small-scale stochastic components.**

Equation 3: G1=Gn*3.3 (remove parentheses)

**We change the expression in Equation 3.**

Line 315: Vp=7.8 km/s is far too high for oceanic L3

**We agree that the endpoint $V_p$ in L3 is high. We therefore decided to regenerate the model (all parameters) and the associated modeling examples for the sake of consistency of the manuscript.**

Line 326: "subducting volcanic ridges" or "subducting seamounts"

**We correct the sentence.**

Line 339: What do you mean by "second-order parameters"? Which are the "first-order" ones?

**We rewrite the sentence. By second-order parameters, we mean those parameters (density, attenuation) that have a small influence on the kinematic of wave propagation (traveltimes) - and therefore on the results of the seismic imaging. In contrast, wavespeed mainly controls traveltimes, hence making their signature in the data dominant for tomography and waveform inversion applications.**

Line 351: Brocher's (2005) is an empirical relationship with significant uncertainty/error bounds. Applying the exact same relationship (same polynomial conversion law) in all units and sectors sounds like too "perfect". In particular this approach "fixes" Vp/Vs. Wouldn't had been better to define and apply slightly modified versions of the conversion laws in different units/sectors, too? Would this have any effect at all on FWI?

**Indeed, the Brocher's compilations are derived from other empirical relations and they are burdened with a certain level of uncertainty. However,**

**since they are expressed as the best-fitting 5th order polynomials, they can generate significant variations in $V_p/V_s$ and $V_p/rho$ ratio from one unit to the next, which is far more realistic than using constant ratios. While it would have been even more realistic to introduce additional deviations (random or geology based) to these relations, they would not have had significant impact on the results of the FWI. A notable exception is however the subduction channel where on top of the Brocher's relations we implement additional small-scale variations of the elastic effects to represent fluid overpressure, fluid diffusion and dry zone according to the interpretation of a migrated section across the Gulf of Guayaquil.**

Line 352: "on laboratory measurements, (...)"

**We correct the sentence.**

Line 366: thermal effects, too?

**We add this suggestion to the list.**

Line 371: Same as in Vs(Vp): wouldn't it be better to allow for a range of variation in the shape of conversion laws to allow for heterogeneous Qp/Qs?

**As we mention before, although this would lead to a more realistic model in terms of parametrisation, we do not expect it to impact the waveform inversion. In addition, the estimation of $Q_p$ and $Q_s$ from the field data is uncertain itself, and therefore adding small deviations to the conversion**

[Figure]

**laws which we apply can still produce a $Q$ model which falls into this uncertainty range and provide the negligible changes into the wavefield generated in this model. By negligible we mean the changes which are too small to influence the large-scale 3D waveform inversion at the current stage of development.**

Line 390: Small-scale perturbations: Interesting approach, it makes all sense introducing small-scale perturbations, although the selection of the size and shape of the SEs seems rather arbitrary in some cases. On the other hand, couldn't it happen that in some instances the values of the stacked SEs align so that the size of the magnitude is larger than 1? I mean up to 4 or -4 if you are stacking four SEs? Or you re-normalize between (-1,1) after stack?

**We use the primitive disk-shaped SE with variable magnitude, since it is easy to control its spatial-scale ratios. This gives a certain level of control on the size and the shape of the final small-scale perturbations. Alternative approach could employ, for example, fractal functions although we did not investigated such an implementation in 3D. Indeed after the stacking the magnitude of the perturbations significantly exceeds the (-1,1) interval. This is because the SEs strongly overlap with each other and for a given SE in the 3D space we have 26 nearest neighbour SEs. Additionally for different structural units we use the sum of the stacks resulting from stacking of the SEs of different scales. Therefore before using the final stochastic matrices to apply the small-scale perturbations we re-normalise the full 3D stack between -1 and 1. We add this information into the body of the manuscript.**

Line 427: "The overall distribution of the energy added to the background medium (...)

[Figure]

is close to normal" > It would be nice to illustrate this with a figure.

> **We add the inset into the Figure 6b presenting the normalised histogram
> of the introduced velocity perturbations. We also rewrite the sentence
> referring to the energy distribution inside the wavenumber spectrum.**

Line 443: How are warping matrices in figs 7a and 7b build?

> **To obtain those matrices, we first generate the matrix of the same size with
> random values. In the second step, we interpolate between the uniformly
> sub-sampled elements of this random matrix using splines. The spatial
> scale of the final shifts in both matrices in Figure 7a and 7b is controlled by
> the sub-sampling - namely the dense/sparse sampling leads to small/large
> scale of perturbations. We add this information to the manuscript.**

General comment: I wonder whether it is necessary that you give precise information
on the expressions used at each step of the process (projection, gradient, physical
parameters, stochastic perturbations, warping) applied at each unit, etc. I mean, if you
do not do this, your results (in this case, the model) are not fully reproducible. At least
not with the information provided.

> **Before we started thinking about this project, we identified the lack of
> such a geomodel in the geophysical imaging community. Our idea was to
> fill this gap and freely release GO_3D_OBS as a benchmark to potential
> users. Through this, we wanted to provide a tool that helps understanding
> better the potential and limits of high-resolution seismic imaging methods
> at the crustal scale and hence stimulate the geophysical community to**

**apply more routinely these techniques. Importantly, this benchmark gives the opportunity to compare the results of different imaging approaches provided the model remains unchanged and available in the form that we present here. As an illustration, the purpose of the synthetic models routinely used in exploration geophysics is not to reproduce or modify them but to provide benchmark for comparing different techniques. Therefore, the ultimate goal of this study is not to discuss a workflows for geomodel building (which is here developed from scratch and would require a tremendous amount of supplementary materials, manuals and data) but to propose a benchmark devoted to the assessment of seismic imaging techniques.**

**Moreover, some components of the model, which are based on the large randomly generated matrices, prevent to readily build the geomodel without access to the heavy amount of intermediate data, which are used as I/O during subsequent steps of the geomodel building. We want to mention here, that due to the size of the files containing the model itself, we encountered problems in finding open-access data-repository, and therefore providing even more data is beyond our abilities.**

**On the other hand, we believe that the different steps implemented to build the geomodel from scratch are described in enough details to build similar models and/or inspire future studies on geomodel building.**

Figure 9: The figure is excellent, although I do not think that it is the best way to show the effect of variable attenuation. It would probably help showing a few individual traces (seismograms) showing details of the effect for different sectors of the model, offsets and recording times. It would help visualizing not only amplitude but also phase differences.

Figure 10: Same as with fig 9. Showing a few well-chosen traces (additionally or alternatively to the whole records in this fig) could also help visualizing differences and

effects. Comment also valid for fig 11m where differences are even slighter.

Line 530: "modelling example, which (...)" or "modelling example that (...)" Figure 12: Same comment as in the previous two figures concerning comparison of several individual traces.

**We correct the sentence in Line 530. Regarding Figures 9-12 presenting different waveform modeling scenarios, we decided to keep our way of data comparison. We understand and appreciate the suggestion about the direct comparison of individual traces recorded at different location of the model. Such an analysis would certainly provide a more precise insight on the footprint of different approximations on wave propagation and on the subsequent inversion. This would however deserve a complete and exhaustive study and a detailed description, which we believe is beyond the main scope of this manuscript. Our aim here was to present an overview of different factors or approximations that can affect imaging techniques (related to the physics, the 2D approximation in complex media or the modeling scheme) and to check to first order that our benchmark can be used to reproduce these effects.**

Line 577-578: An extra question to be considered: what can be gained from joint inversion of spatially coincident OBS and MCS data? Would it somehow mitigate the need of "densely sampled" OBS acquisitions?

**Thank you for reminding us about the benefit of coincident MCS+OBS acquisitions. Indeed, combining OBS and long-streamer MCS acquisition still remain beneficial since they provide images of the crust at different (complementary) scales and depths, in particular because it remains challenging to push FWI at very high frequencies (beyond 15Hz) due to**

computational cost and error accumulation during the nonlinear iterations. From more methodological viewpoints, the final FWI model from the OBS data can be used as an initial model for the subsequent FWI of the MCS data to further increase its resolution at shallow and intermediate depths (taking advantage of the higher frequency content and the higher fold of the MCS data relative to OBS counterpart). Therefore, the shallow FWI model inferred from MCS data can substitute the FWI model inferred from OBS data in shallow areas when the latter is polluted by aliasing artefacts resulting from coarsely sampled OBSs.

Also, the MCS data can be migrated with the velocities estimated by FWI to tentatively image reflectors at depths where migration-based velocity analysis are ineffective due to insufficient reflection move-out, hence prolongating at greater depths the depth-migrated images (Gorszczyk et al. 2019). Indeed, performing 3D towed-streamer surveys in Academia seems out of range due to the lack of equipment and the cost of these surveys (in particular, at the scale of a margin). Only, a coarse grid of MCS lines can be viewed today, which remain highly beneficial and complementary to 3D OBS acquisition.

As the OBS deployment geometry, the shooting strategy during 3D OBS experiments will need to be optimized to maintain reasonable acquisition time while optimizing imaging resolution. Definitively, the GO_3D_OBS geomodel should help to optimize the design of the next generation 3D academic surveys considering the limited pools of OBS and the limited acquisition time made available to the academic marine geophysics community.

Line 590: Uncertainty estimation: While it is true that uncertainty analysis has commonly been overlooked, it is becoming more and more common in recent times. Several schemes have been proposed and it is now routinely done in many travel-time

tomography studies. As an example, the description of a formal Monte Carlo sampling scheme-based analysis can be found in Korenaga & Sager (2012). I'd say that the actual situation deserves a reference in this section.

*Korenaga, J. & Sager, W.W., 2012. Seismic tomography of Shatsky Rise by adaptive importance sampling, J. geophys. Res., 117, B08102, doi:10.1029/2012JB009248.*

**Thank you for mentioning this reference. We extend the Uncertainty estimation section in the updated manuscript.**
* * *

---

## Author Comment (AC4) · 25 Jan 2021

Rie Nakata

The manuscript describes about novel model development efforts based on the authors' previous works in the eastern Nankai Trough. The paper is well written and mostly easy to follow. The developed model will be useful for the community including the model developing methodologies. I found two components are missing:

[Figure]

i) comparisons of waveforms to observed ones : are the waveforms representative enough? , ii) comparisons with drilling efforts or onshore proxy sites for physical properties. Some discussions will be useful.

**Dear Rie,**

**Thank you for your positive assessment of our work and for your constructive comments. Please find hereafter our answers.**

**Best regards**
**The authors**

It is fine to build a model with a focus on multiparameter FWI – but slightly differs from title (imaging). You start FWI as "velocity building" which is typically different from "imaging". You may want to clarify these points and perhaps add "FWI" in abstract. Can the authors discuss if we can simulate 3D reflection dataset (as in Kumano) using the model and test various imaging methods too? Are the grid sizes etc sufficient?

**In the context of this study, imaging should be understood as any procedure for estimating the earth's rock parameters from seismic data - including traveltime tomography, migration-based velocity analysis, migration and full waveform inversion. This term has therefore a broader meaning than what is typically behind the "imaging" term referring to migration-like techniques only, which are mostly used in exploration geophysics. We will put more emphasis on this terminology issue in the manuscript to prevent misunderstanding and distinguish velocity model building techniques from migration techniques in addition to all-at-once**

**approaches as full waveform inversion.**

**Simulating 2D or 3D reflection dataset with a 25m grid interval to test migration techniques is of course possible. In fact, we already did some tests in 2D using a profile of the GO_3D_OBS geomodel (AGU abstract Sambolian(2019), SEG abstract Alashloo(2020)). On the original grid (25 m) with acoustic approximation, accurate and stable modeling is possible up to frequencies 15 Hz - 20 Hz (depending on the accuracy order of the stencil). To allow higher frequency content or elastic modeling, one may need to resample the model on a finer grid. We decided to define the original model version using 25 m grid to avoid extremely large volume of the files containing the model (132 GB per parameter using 25 m grid)**

There are numerous drilling efforts in the subduction zones and field sampling in proxy sites, including those off the Kumano-nada region of the Nankai trough. The physical properties (and so on) should reflect the results. Please add comments on how your model leverage these efforts.

Eastern Nankai?: Kingston's work is off Kumano and thus not eastern Nankai. As your structural model significantly depends on his work, I suggest to remove specific reference to "eastern" and add references of Kumano too.

The authors describe very lightly about applicability to other subduction zone studies. How much does the model applicable and what sense? Is the model applicable to erosional margins? Or the procedure used to build a model? Adding more references is also important.

**We agree that drilling provides useful information about the physical properties of the subsurface. This information, however, can rapidly change between nearby drilling sites across the same margin (for example a single segment of the Nankai Trough) and is mainly shallow (down to**

∼1 km below the sea floor). Moreover, the resolution of the drilling logs is much higher compared to the scale of the structures that we incorporate in the model.

As mentioned by the second reviewer, we were referring too specifically to the eastern-Nankai Trough region in the original manuscript, while our overall goal is to provide a representative complex crustal-scale bench-mark model of subduction zones by gathering useful information from different areas (while maintaining geological consistency). Therefore, we want to stress that our goal is not to build a geomodel of a specific area.

Accordingly, we indirectly take benefit from drilling information through the Brocher's relations, which gather informations from different geological environments coming from either numerous field or laboratory measurements. This allows us to combine geological features from different subduction zones (including Kumano accretionary prism interpreted by Kingston - we will mention Kumano-nada region in the manuscript) with the aim to make the structure as realistic and complex as possible. However, the scales and shapes of those structures are necessarily modified and therefore they cannot represent accurately the subduction zone of a specific area.

At the beginning of Section 2.1 *Geological features* we write:

"*The overall geological setup of our model is mainly (but not only) inspired by the features interpreted in the Nankai Trough area. However, these structures can be also found in different margins around the world combined in various configurations. Therefore, our model is not intended to replicate a particular subduction zone and its related geology for geodynamic studies of the targeted region. On the contrary, it was designed to comprise broad features one may encounter in these tectonic environments.*"

The purpose of this study is therefore to provide a realistic crustal-scale

**geomodel to test any seismic imaging methods that may be useful at this scale (tomography, any kind of migration, FWI) and related issues as survey design, high-performance computing issues etc. etc. We choose subduction zone for its geological complexity, the associated variations of the physical parameters, and the fundamental issues related to the better understanding of the structural factors controlling the rupture process of a megathrust earthquake at seismogenic zones. Recent imaging works performed by ourselves in this setting provide us the initial guidelines and inspiration to perform this study. We afraid that it might inappropriate to study a particular erosional margin with our geomodel, although some other inferences regarding e.g. resolution analysis of FWI from wide-angle OBS data or crosstalk between parameters during multi-parameter inversion, might remain valid to first order. Indeed, nothing prevents building such an erosional margin model or any other model for specific studies using the approaches we presented in the manuscript.**

are there specific problems encountered in previous FWI/imaging works apart of scaling issues? For example, as seen in Park et al (2010) and addressed in Kamei et al. (2012), low velocity zones were problematic for MVA and subsequent earthquake fault imaging was really a problem (perhaps blank accretionary prisms) – the authors rightly mention "trapped" waves etc. Adding imaging/inversion issues for (specific) important geological features will be helpful (and if they are incorporated into).
"deep" targets: what do you mean by deep targets? Perhaps you can be more specific?

**Of course, the dimension of the target is one issue - especially from the high-performance computing (HPC) viewpoint (indeed, each user can extract any target from the full model to focus on specific areas of the**

**model). In the manuscript, we review several methodological issues
that could be investigated with our model: optimal survey design and
sparsity-promoting regularisation to deal with sparse acquisitions; ini-
tial velocity model building for FWI; detrimental effects of out of plane
wavefield propagation during 2D imaging; nonlinearity of the FWI and
design of robust misfit function to mitigate cycle skipping, resolution
analysis, multi-parameter imaging, ... On the structural side, we try to
incorporate most of the geological features at different scales that have
been documented in subduction zones, including underplating of crustal
sheets, complex thrusts and folds in the accretionary wedge, thrusts with
damaged zones, steep and mild faults in the subducting oceanic crust,
sedimentary basins, thin subduction channel with heterogeneous lateral
properties, duplex, ridges etc. Those structures, which should generate
wavefields whose anatomy is similar to those recorded in the field, can
raise different challenges depending on the applied technique, acquisition
design and physics approximation.**

**Thank you for bringing our attention to "deep targets". We shall be
more specific. By deep we mean the targets which cannot be precisely
reconstructed using the data acquired with typical streamer length - ∼6
km. We edit the manuscript to be more precise.**

Great to see a range of different modeling efforts. Some motivational statements
will be helpful: Why do these modeling important? Are you recommending to
generate 3D (visco-)elastic spectral-element waveforms and apply a method of
imaging/inversion? Are you going to make these waveforms available as "datasets" as
done by BP/Chevron etc?

**Thank you for this suggestion. We will underline the importance of**

**different modeling approximation.**

**In Section 4.5 *Further development* we mention:**

*"On the other hand, generating accompanying dataset for the current models can further broaden its impact. Possible dataset could include sparse 3D and dense 2D OBS deployments, as well as corresponding streamer data. Such dataset could be directly use for a given type of processing."*

**Therefore, it is true that we plan to release such open-access datasets in the future. We already did some tests with 2D OBS and 2D MCS data - including modeling, depth-migration and acoustics, visco-acoustic and elastic FWI techniques. We need to check very carefully the accuracy of the wavefield simulations for different physics before making datasets, preferably OBS and MCS, available for the community. We need also to define several representative targeted area (2D and 3D) in terms of location and dimension such that a user can select the most suitable one for his study (which can be geologically- or methodologically-driven).**

Interwoven OBS gathers are nice, but some of the authors descriptions are difficult to follow (esp. 2D vs 2.5D vs 3D) unless scrutinizing those plots. The authors should add arrows (e.g. Pn waves or representative off-planer waves). Also it would be easier to understand "complexity" if the authors show 2.5D snapshots along with 3D snapshots. Please add amplitude spectrum and amplitude-vs-offset curves to show the spectra esp. for pure vs visco acoustic simulations to quantitatively display the discrepancies. Please add comments on whether/how much the modeled waveforms represent the observed waveforms (e.g. in eastern Nankai) to convince the model is representative. A word for choosing spectral element at the start of section 3.3 will be beneficial rather than at the end.

**We augment the figures to make them more exhaustive. We also provide the insight into example of the field OBS gather.**
**Considering all of the pros and cons of the spectral element modeling engine, we decide to use the SEM46 code developed in the framework of the SEISCOPE project since it allows us to perform elastic modeling in marine environment with a high accuracy and an adaptive mesh, while avoiding the detrimental staircase effects of the finite-difference method at the sea bottom.**

The authors discuss about benefits to FWI/tomography/acquisition. How does the model help imaging? How the model help bridging imaging and tomography gaps?

**In the Introduction, we mention how high resolution velocity reconstruction methods like FWI applied to the OBS data can produce the background velocity models for the migration of the MCS reflection data. Moreover, the high-resolution FWI models can be jointly interpreted with the reflectivity section making the interpretation more valid (Górszczyk et al. 2019). Therefore, establishing robust FWI approaches for the processing of OBS data using synthetic tests can mitigate the resolution gap between tomography and migration.**

small scale perturbations "disk-shaped structural elements": what do these who geologically? What do they need to overlap?

**We use the primitive disk-shaped SE with variable magnitude, since it is easy to control its spatial-scale ratios. This gives a certain level of control on the size and the shape of the final small-scale perturbations. They**

**overlap to avoid introduction of artificially looking disc shaped anomalies. Through their dense spatial positioning and stacking, we obtain a random/noisy background perturbations as presented in Figure 6a.**

Figure 1: Is the figure necessary? I do not know if any outside FWI community understands the figure without further expanding the descriptions. The manuscript is about model not FWI. I think the figure is unnecessary.

**We agree that Figure 1 might be addressed to the FWI expert. However, since the manuscript is also about how to use the model to assess methods and design survey geometry, we believe it makes sense to mention the theoretical guidelines that may be followed to perform this assessment. Therefore, we keep the Figure 1 as it is.**

Figure 2: Add meanings of the lines in Figure 2a in the caption.

**We edit the caption.**

Figure 5: Perhaps add a vertical profile?

**We present vertical inline profiles.**

Figure 6: c-f: what are numbers on the top left? "Red/blue colours indicate: : :" is difficult to follow.

**We edit the caption.**

RMS: is not defined.

**We explain the abbreviation.**
* * *